# Bulk tissue cell type deconvolution with multi-subject single-cell expression reference

Xuran Wang[1], Jihwan Park [2], Katalin Susztak[2], Nancy R. Zhang [3] & Mingyao Li[4]

Knowledge of cell type composition in disease relevant tissues is an important step towards the identification of cellular targets of disease. We present MuSiC, a method that utilizes cell-type specific gene expression from single-cell RNA sequencing (RNA-seq) data to characterize cell type compositions from bulk RNA-seq data in complex tissues. By appropriate weighting of genes showing cross-subject and cross-cell consistency, MuSiC enables the transfer of cell type-specific gene expression information from one dataset to another. When applied to pancreatic islet and whole kidney expression data in human, mouse, and rats, MuSiC outperformed existing methods, especially for tissues with closely related cell types. MuSiC enables the characterization of cellular heterogeneity of complex tissues for understanding of disease mechanisms. As bulk tissue data are more easily accessible than single-cell RNA-seq, MuSiC allows the utilization of the vast amounts of disease relevant bulk tissue RNA-seq data for elucidating cell type contributions in disease.

[1] Graduate Group in Applied Mathematics and Computational Science, University of Pennsylvania, Philadelphia, PA 19104, USA. [2] Departments of Medicine and Genetics, University of Pennsylvania, Philadelphia, PA 19104, USA. [3] Department of Statistics, The Wharton School, University of Pennsylvania, Philadelphia, PA 19104, USA. [4] Department of Biostatistics, Epidemiology & Informatics, University of Pennsylvania, Philadelphia, PA 19104, USA. Correspondence and requests for materials should be addressed to N.R.Z. (email: nzh@wharton.upenn.edu) or to M.L. (email: mingyao@pennmedicine.upenn.edu)

Bulk tissue RNA-seq is a widely adopted method to understand genome-wide transcriptomic variations in different conditions such as disease states. Bulk RNA-seq measures the average expression of genes, which is the sum of cell type-specific gene expression weighted by cell type proportions. Knowledge of cell type composition and their proportions in intact tissues is important, because certain cell types are more vulnerable for disease than others. Characterizing the variation of cell type composition across subjects can identify cellular targets of disease, and adjusting for these variations can clarify downstream analysis.

The rapid development of single-cell RNA-seq (scRNA-seq) technologies have enabled cell type-specific transcriptome profiling. Although cell type composition and proportions are obtainable from scRNA-seq, scRNA-seq is still costly, prohibiting its application in clinical studies that involve a large number of subjects. Furthermore, scRNA-seq is not well suited to characterizing cell type proportions in a solid tissue, because the cell dissociation step is biased towards certain cell types[1].

Computational methods have been developed to deconvolve cell type proportions using cell type-specific gene expression references[2]. CIBERSORT[3], based on support vector regression, is a widely used method designed for microarray data. More recently, BSEQ-sc[4] extended CIBERSORT to allow the use of scRNA-seq gene expression as a reference. TIMER[5], developed for cancer data, focuses on the quantification of immune cell infiltration. These methods rely on pre-selected cell type-specific marker genes, and thus are sensitive to the choice of significance threshold. More importantly, these methods ignore cross-subject heterogeneity in cell type-specific gene expression as well as within-cell type stochasticity of single-cell gene expression, both

of which cannot be ignored based on our analysis of multiple scRNA-seq datasets (Supplementary Figure 1a).

Here we introduce a new MUlti-Subject SIngle Cell deconvolution (MuSiC) method (code available) that utilizes cross-subject scRNA-seq to estimate cell type proportions in bulk RNA-seq data. Through comprehensive benchmark evaluations, and applications to pancreatic islet and whole kidney expression data in human, mouse, and rats, we show that MuSiC outperformed existing methods, especially for tissues with closely related cell types.

## Results

**Methods overview.** An overview of MuSiC is shown in Fig. 1. MuSiC starts with multi-subject scRNA-seq data, and assumes that the cells for each subject have been classified into a set of fixed cell types that are shared across subjects. MuSiC deconvolves bulk RNA-seq samples to obtain the proportions of these cell types in each sample. A key concept in MuSiC is "marker gene consistency". We show that, when using scRNA-seq data as a reference for cell type deconvolution, two fundamental types of consistency must be considered: cross-subject and cross-cell, in which the first is to guard against bias in subject selection, and the second is to guard against bias in cell capture in scRNA-seq. By incorporating both types of consistency, MuSiC allows for scRNA-seq datasets to serve as effective references for independent bulk RNA-seq datasets involving different individuals.

Rather than pre-selecting marker genes from scRNA-seq based only on mean expression, MuSiC gives weight to each gene, allowing for the use of a larger set of genes in deconvolution. The weighting scheme prioritizes consistent genes across subjects:

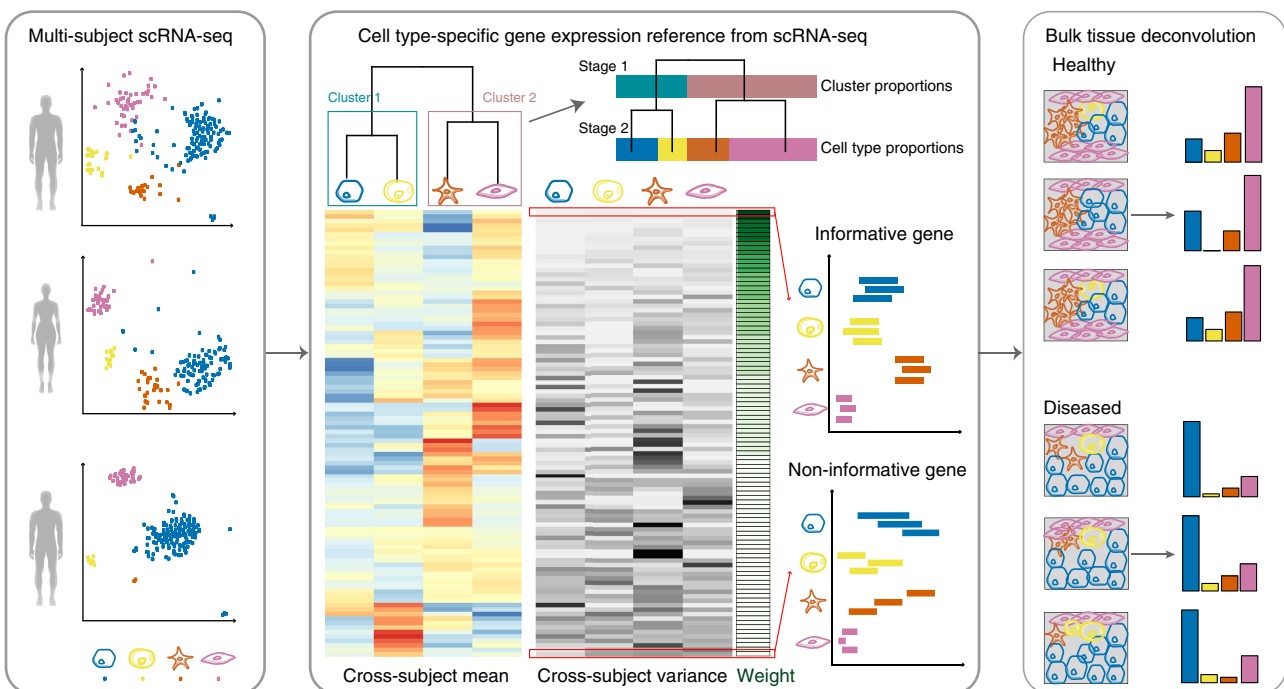

**Fig. 1** Overview of MuSiC framework. MuSiC starts from scRNA-seq data from multiple subjects, classified into cell types (shown in different colors), and constructs a hierarchical clustering tree reflecting the similarity between cell types. Based on this tree, the user can determine the stages of recursive estimation and which cell types to group together at each stage. MuSiC then determines the group-consistent genes and calculates cross-subject mean (red to blue) and cross-subject variance (black to white) for these genes in each cell type. MuSiC up-weighs genes with low cross-subject variance and down-weighs genes with high cross-subject variance. In the example shown, deconvolution is performed in two stages, only cluster proportions are estimated for the first stage. Constrained by these cluster proportions, the second stage estimates cell type proportions, illustrated by the length of the bar with different colors. The deconvolved cell type proportions can then be compared across disease cohorts

up-weighing genes with low cross-subject variance (informative genes) and down-weighing genes with high cross-subject variance (non-informative genes). This requirement on cross-subject consistency is critical for transferring cell type-specific gene expression information from one dataset to another.

Solid tissues often contain closely related cell types, and correlation of gene expression between these cell types leads to collinearity, making it difficult to resolve their relative proportions in bulk data. To deal with collinearity, MuSiC employs a tree-guided procedure that recursively zooms in on closely related cell types. Briefly, we first group similar cell types into the same cluster and estimate cluster proportions, then recursively repeat this procedure within each cluster (Fig. 1). At each recursion stage, we only use genes that have low within-cluster variance, a.k.a. the cross-cell consistent genes. This is critical as the mean expression estimates of genes with high variance are affected by the pervasive bias in cell capture of scRNA-seq experiments, and thus cannot serve as reliable reference. See online methods for details.

**Application to pancreatic islets in human**. To demonstrate and evaluate MuSiC, we started with a well-studied tissue, the islets of Langerhans, which are clusters of endocrine cells within the pancreas that are essential for blood glucose homeostasis. Pancreatic islets contain five endocrine cell types ($\alpha, \beta, \delta, \epsilon$, and $\gamma$), of which $\beta$ cells, which secrete insulin, are gradually lost during type 2 diabetes (T2D). We applied MuSiC to bulk pancreatic islet RNA-seq samples from 89 donors from Fadista et al.[6], to estimate cell type proportions and to characterize their associations with hemoglobin A1c (HbA1c) level, an important biomarker for T2D. We were motivated to re-analyze this data because, as shown in Fig. 2 and in Baron et al.[4], existing methods failed to recover the correct $\beta$ cell proportions, which should be around 50–60%[7], and also failed to recover their expected negative relationship with HbA1c level. As reference, we experimented with scRNA-seq data from two sources: 6 healthy and 4 T2D adult donors from Segerstolpe et al.[8], and 12 healthy and 6 T2D adult donors from Xin et al.[9]. All bulk and single-cell datasets in this analysis are summarized in Table 1.

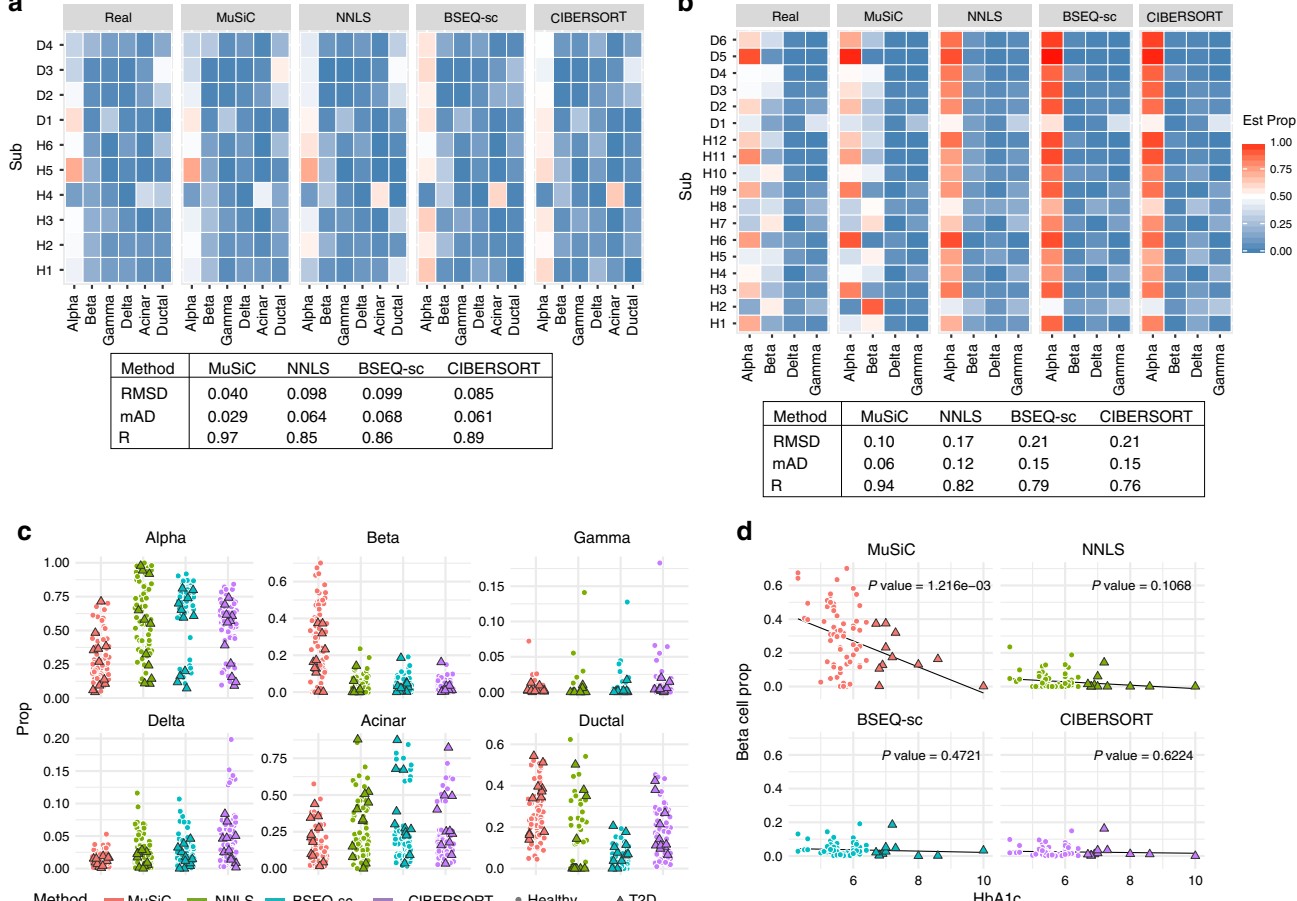

**Fig. 2** Pancreatic islet cell type composition in healthy and T2D human samples. **a, b** Benchmarking of deconvolution accuracy on bulk data constructed by combining together scRNA-seq samples. **a** The bulk data is constructed for 10 subjects from Segerstolpe et al. while the single-cell reference is taken from the same dataset. The cell type proportions of healthy subjects are estimated by leave-one-out single cell reference. The subject names are relabeled; the table shows average root mean square error (RMSD), mean absolute deviation (mAD), and Pearson correlation (R) across all samples and cell types. **b** The bulk data is constructed for 18 subjects from Xin et al. while the single cell reference is six healthy subjects from Segerstolpe et al. **c** Jitter plots of estimated cell type proportions for Fadista et al. subjects, color-coded by deconvolution method. Of the 89 subjects from Fadista et al., only the 77 that have recorded HbA1c level are plotted, and T2D subjects are denoted as triangles while non-diabetic subjects are denoted as dots. **d** HbA1c vs beta cell type proportions estimated by each of 4 methods. The reported *p*-values are from single variable regression $\beta$ cell proportion ~HbA1c. Multivariable regression results are reported in Supplementary Table 1. Supplementary Figure 7 shows the deconvolution results of Fadista et al. with the inDrop data from Baron et al. as single-cell reference. The corresponding multivariable regression results are shown in Supplementary Table 2. Source data are provided as a Source Data file

**Table 1 Pancreatic islet datasets**

| Name | Journal | Year | Accession # | Tissue type | Data type | Protocol | # Samples | # Cells | # Genes | # Cell types |
|---|---|---|---|---|---|---|---|---|---|---|
| Segerstolpe et al.[8] | Cell Metabolism | 2016 | E-MTAB-5061 | Pancreatic islet | Single-cell RNA-seq | Smart-seq2 | 10 (6 H + 4 T2D) | 2209 | 25,453 | 14 + 1 NA |
| Segerstolpe et al.[8] | Cell Metabolism | 2016 | E-MTAB-5060 | Pancreatic islet | Bulk RNA-seq | Smart-seq2 | 7 (3 H + 4 T2D) | NA | 25,453 | NA |
| Xin et al.[9] | Cell Metabolism | 2016 | GSE81608 | Islet: endocrine | Single-cell RNA-seq | Illumina HiSeq 2500 | 18 (12 H + 6 T2D) | 1492 | 39,849 | 4 |
| Baron et al.[4] | Cell Systems | 2016 | GSE81433 | Pancreatic islet | Single-cell RNA-seq | Illumina HiSeq 2500 (InDrop) | 3 healthy | 7729 | 17,434 | 14 + 1 NA |
| Fadista et al.[6] | PNAS | 2014 | GSE50244 | Pancreatic islet | Bulk RNA-seq | Illumina HiSeq 2000 | 89 | NA | 56,638 | NA |

First, to systematically benchmark, we applied MuSiC and three other methods (Nonnegative least squares (NNLS), CIBERSORT, and BSEQ-sc) to artificial bulk RNA-seq data constructed by simply summing the scRNA-seq read counts across cells for each single-cell sequenced subject. In this case, true cell type proportions are known, which allows the evaluation of accuracy. More details on artificial bulk construction are described in the Supplementary Note 1. Figure 2a, Supplementary Figures 1c and 2b show the estimation results when the artificial bulk and the single-cell reference data are from the same study, either both from Segerstolpe et al.[8] or both from Xin et al.[9]. MuSiC achieves improved accuracy over existing procedures. Figure 2b and Supplementary Figure 2a show the estimation results when the artificial bulk and the single-cell reference data are from different studies. This is a more challenging but more realistic scenario, since library preparation protocols vary across labs and bulk deconvolution analyses are often performed using single-cell reference generated by others. MuSiC still maintains high accuracy, while other methods perform substantially worse. Further comparisons show that, unlike existing methods that rely on pre-selected marker genes, MuSiC gives accurate results when the cell type composition in the bulk data is substantially different from that of the single cell reference (Supplementary Figure 2c and Supplementary Note 2), and when the bulk tissue contains minority cell types that are missing in the reference (Supplementary Figure 3 and Supplementary Note 3). MuSiC's ability to transfer knowledge across data sources is derived from its consideration of marker gene consistency.

We now turn to the deconvolution of bulk RNA-seq data from Fadista et al.[6]. We first used the scRNA-seq data from Segerstolpe et al. as reference for all methods. MuSiC recovers the expected ~50–60% β cell proportion for the healthy subjects[7], whereas other methods grossly overestimate the proportion of α cells and underestimate the proportion of β cells. Furthermore, MuSiC detects a significant association of β cell proportion with HbA1c level (p-value 0.00126, Fig. 2d). Based on clinical standard, HbA1c level <6.0% is classified as normal, and >6.5% is classified as diabetic. After adjusting for age, gender and body mass index, MuSiC estimates suggest that a 0.5% increase in HbA1c level, representing the magnitude of increase from normal to the diabetes cutoff, corresponds to a drop of 3.07% ± 2.49% in β cell proportion (Supplementary Table 1). The scRNA-seq data from Segerstolpe et al. was generated by the Smart-seq2 protocol. Similar results are obtained when using the InDrop scRNA-seq data from Baron et al. as reference. MuSiC detects the significant association of β cell proportion with HbA1c level with and without adjustment for covariates (Supplementary Figure 7, Supplementary Table 2). The weight ordered gene list for pancreatic islet analysis are provided in Supplementary Table 5.

**Application to kidney in mouse and rats**. As a second tissue example, we used the kidney, a complex organ consisting of several anatomically distinct segments each playing critical roles in the filtration and reabsorption of electrolytes and small molecules of the blood. Chronic kidney disease (CKD), the gradual loss of kidney function, is increasingly recognized as a major health problem, affecting 10–16% of the global adult population. We aim to characterize how kidney cell type composition changes during CKD. Fibrosis is the histologic hallmark common to all CKD models, and hence, we analyzed the bulk RNA-seq data from three mouse models for renal fibrosis: unilateral ureteric obstruction induced by surgical ligation of the ureter (UUO, Arvaniti et al.[10]), toxic precipitation in the tubules induced by high dose folic acid injection (FA, Craciun et al.[11]), or genetic alteration by transgenic expression of genetic risk variant APOL1 in podocytes (APOL1 transgenic mice[12]). As reference, we used the mouse kidney-specific scRNA-seq data from Park et al.[1]. Details of all datasets are summarized in Table 2. We systematically benchmarked all methods on artificial bulk experiments performed using the Park et al. scRNA-seq data, finding similar trends as those in Fig. 2a, b (Supplementary Figure 4a, b).

Hierarchical clustering of the cell types in the single cell reference reveals that, apart from neutrophils and podocytes, kidney cells fall into two large groups: Immune cell types (macrophages, fibroblasts, T lymphocytes, B lymphocytes, and natural killer cells) and kidney-specific cell types (proximal tubule (PT), distal convoluted tubule, loop of Henle, two cell types forming the collecting ducts, and endothelial cells). Of these, PT is the dominant cell type in kidney, and the proportion of PT cells is known to decrease with CKD progression. MuSiC finds this decrease in all three mouse models (Fig. 3b–d). Other methods also detect this association for the APOL1 and UUO mouse models, but showed ambiguous results for the FA model.

Distal convoluted tubule cells (DCT) are known to be the second most numerous cell type in kidney, with an expected proportion of ~10–20%[1]. Yet, CIBERSORT did not detect DCT in any of the three bulk datasets; BSEQ-sc missed it in two datasets and grossly over-estimated its proportion in the third dataset at the cost of a grossly underestimated PT proportion. This is due to the high similarity between DCT and PT, observable in Fig. 3a. Through its tree-guided recursive algorithm, MuSiC first estimates the combined proportion of kidney cell types versus immune cell types using consistent genes for these two large groups, and then zooms in and deconvolves the kidney cell types using genes re-selected for each kidney cell type. This allows MuSiC to successfully separate PT and DCT cells in all three bulk datasets, recovering a consistent DCT proportion between 8–20%, matching expectations. Interestingly, unlike for PT, the proportion of DCT cells show a consistent increase with

**Table 2 Mouse/rat kidney datasets**

| Name | Journal | Year | Accesession # | Tissue type | Data type | Protocol | # Samples | # Cells | # Genes | # Cell types |
|---|---|---|---|---|---|---|---|---|---|---|
| Park et al.[1] | Science | 2018 | GSE107585 | Kidney | Single-cell RNA-seq | 10x | 7 health, male | 43,745 | 16,273 | 14 + 2 novel |
| Beckerman et al.[12] | Nature Medicine | 2017 | GSE81492 | Kidney | Bulk RNA-seq | Illumina HiSeq 2500 | 10 (6 control + 4 APOL1) | NA | 19,033 | NA |
| Lee et al.[13] | JASN | 2015 | GSE56743 | Kidney tubule | Bulk RNA-seq | Illumina HiSeq 2000 | 118 replicates (14 segments) | NA | 10,903 | NA |
| Craciun et al.[11] | JASN | 2015 | GSE65267 | Kidney | Bulk RNA-seq | Illumina HiSeq 2000 | 18 replicates (6 time points) | NA | 25,219 | NA |
| Arvaniti et al.[10] | Scientific Reports | 2016 | GSE79443 | Kidney | Bulk RNA-seq | Illumina HiSeq 2000 | 10 replicates (Sham + 2 time points) | NA | 38,683 | NA |

disease progression across all three mouse models. This may seem counterintuitive given that loss of kidney function is expected to be associated with the loss of kidney cell types. But given the substantial drop of the dominant PT cell type, the proportion of DCT cells relative to the whole may increase, even if its absolute count drops.

Next, we consider immune cells, which are known to play a central role in the pathogenesis of CKD. MuSiC found the largest immune sub-type to be macrophage, and all methods detected the expected increase of macrophage proportion with disease progression. Apart from this, MuSiC also found fibroblasts, B-lymphocytes, and T-lymphocytes to increase in proportion with disease progression, giving a consistent immune signature that is reproduced across mouse models. These findings are consistent with clinical and histological observations, indicating tissue inflammation is a consistent feature of kidney fibrosis. Such reproducible signatures were not found by other methods, which show much less agreement across mouse models. The weight ordered gene list for the three mouse models are provided in Supplementary Table 6–7.

Finally, to illustrate MuSiC's cross-species applicability, we used the mouse kidney scRNA-seq reference from Park et al.[1] to deconvolve the micro-dissected segment aggregated rat RNA-seq data from Lee et al.[13], which contains 105 samples obtained from 14 segments spaced along the renal tubule. Cell type proportions are estimated with homologous genes between mice and rat. We mapped samples to their physical locations, and computed correlations between their cell type proportions (Fig. 3e). Reassuringly, cell types recovered by MuSiC for each segment agree with knowledge (Zhai et al.[14]) about the dominant cell type at its mapped position, e.g. DCT cells come from the DCT segment. Correlation between samples is also high within anatomically distinct segments.

**Evaluation of robustness for MuSiC.** A good deconvolution method should be robust to the choice of single-cell reference. We conducted additional experiments to evaluate the robustness of MuSiC and other existing methods. First, we considered the case where cell type proportions in the single cell data are drastically different from those in the bulk data. Our results indicate that, under this scenario, MuSiC recovers the true cell type composition, improving upon the severely biased estimates produced by other existing approaches (Supplementary Note 2, Supplementary Figure 2c). One limitation of scRNA-seq is that it may fail to recover some cell types, in particular, rare cell types may be missed. We next created considered the setting where the single-cell reference is incomplete, and found that MuSiC estimation is still accurate as long as the missing cell type is not the dominant cell type in bulk tissue (Supplementary Note 3, Supplementary

Figure 3, and Supplementary Table 3). MuSiC is also tolerant of different scRNA-seq protocols. This has already been shown through the above analyses, where accurate deconvolution results were obtained using single cell reference generated using the Smart-seq2, inDrop, and 10x Chromium protocols. To probe this further, we directly investigated the impact of using biased values of relative abundance $\theta_{jg}^k$ in MuSiC's deconvolution step, and found that MuSiC estimated cell type proportions remain accurate, still improving upon existing methods, even though unbiased relative abundance values were provided to the existing methods as input (Supplementary Note 4, Supplementary Figure 8c). Finally, we evaluated the impact of dropout in the single cell reference, by introducing dropout according to Jia et al.[15] and varying the dropout rate in the benchmark experiment of Fig. 2. MuSiC estimation is still accurate even when dropout rate is around 30% (Supplementary Note 5, Supplementary Figure 8a–b).

## Discussion

Knowledge of cell type composition in disease relevant tissues is an important step towards the identification of cellular targets in disease. Although most scRNA-seq data do not reflect true cell type proportions in intact tissues, they do provide valuable information on cell type-specific gene expression. Existing cell type deconvolution methods rely on pre-selected marker genes and ignore subject-to-subject variation and cross-cell consistency in gene expression. Through comprehensive benchmark evaluations and analysis of multiple real datasets, we show that both cross-subject and cross-cell consistency in gene expression need to be considered in deconvolution. By incorporating both types of consistency, MuSiC allows for scRNA-seq datasets to serve as effective references for independent bulk RNA-seq datasets involving different individuals. Harnessing multi-subject scRNA-seq reference data, MuSiC reliably estimates cell type proportions from bulk RNA-seq, therefore enabling the transfer of cell type-specific gene expression from one dataset to another. As bulk tissue data are more easily accessible than scRNA-seq, MuSiC allows the utilization of the vast amounts of disease relevant bulk tissue RNA-seq data for elucidating cell type contributions in disease.

Although this paper uses read counts as the measures of mRNA abundance, there are many other commonly used measures, such as RPKM and TPM. MuSiC can utilize RPKM if estimates of cell type specific total RNA abundance can be provided (e.g., estimated from another data set). However, cell type proportions cannot be estimated with TPM as the input. Detailed interconversion between read counts and other gene expression measures is discussed in Methods.

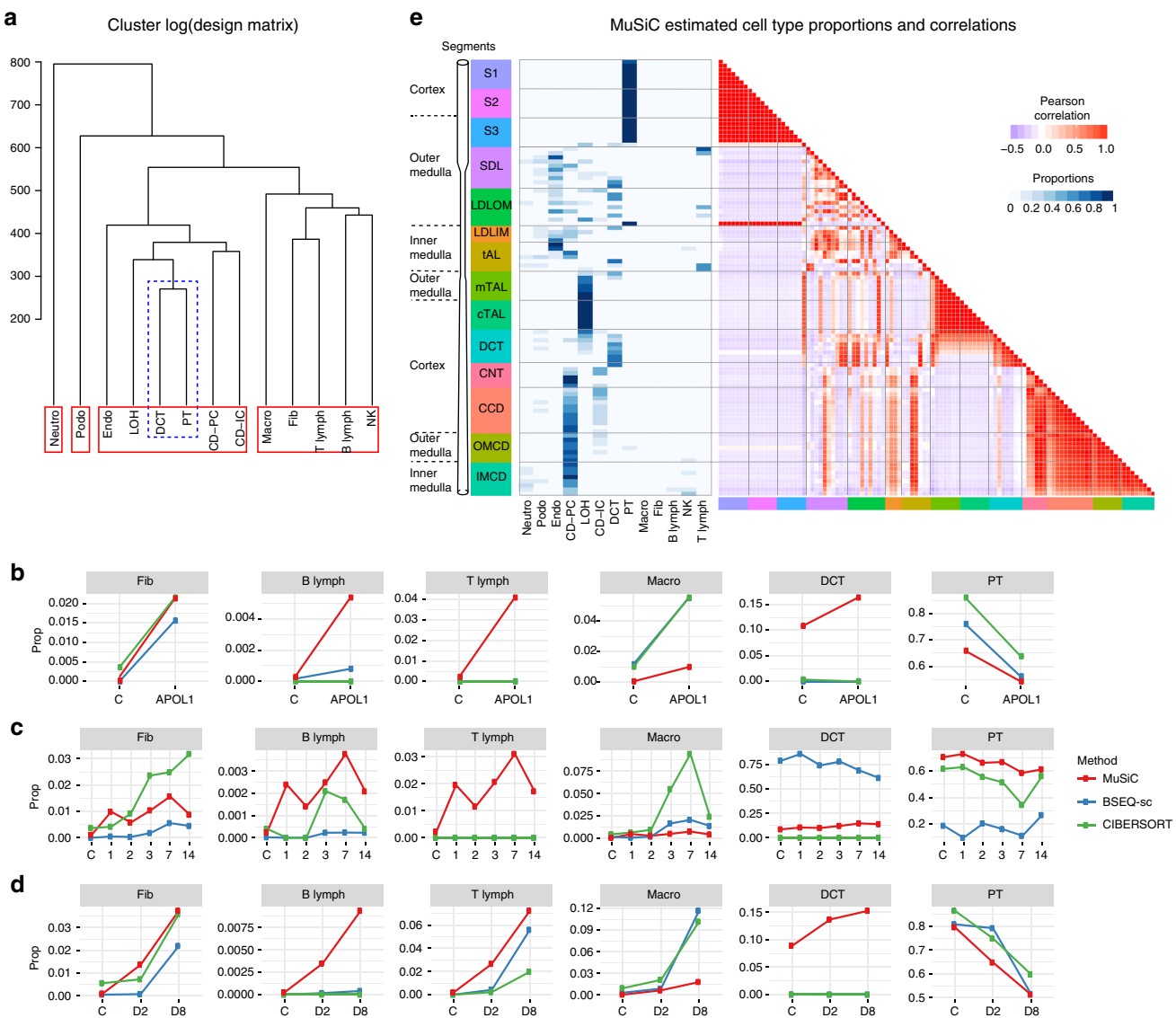

**Fig. 3** Cell type composition in kidney of mouse CKD models and rat. **a** Cluster dendrogram showing similarity between 13 cell types that were confidently characterized in Park et al. Abbreviations: Neutro: neutrophils, Podo: podocytes, Endo: endothelials, LOH: loop of Henle, DCT: distal convoluted tubule, PT: proximal tubule, CD-PT: collecting duct principal cell, CD-IC: CD intercalated cell, Macro: macrophages, Fib: fibroblasts, NK: natural killers. **b–d** Average estimated proportions for 6 cell types in bulk RNA-seq samples taken from three different studies, each study based on a different mouse model for chronic kidney disease. Results from three different deconvolution methods (MuSiC, BSEQ-sc and CIBERSORT) are shown by different colors. Supplementary Figure 5a–c show complete estimation results of all 13 cell types. **b** Bulk samples are from Beckerman et al., who sequenced 6 control and 4 APOL1 mice. **c** Bulk data are from Craciun et al.[9], where samples are taken before (C) and at 1, 2, 3, 7, 14 days after administering folic acid. Line plot shows cell type proportion changes over time (days), averaged over 3 replicates at each time point. **d** Bulk data are from Arvaniti et al.[10], where samples are taken from mice after Sham operation (C), 2 days after UUO operation (D2), and 8 days after UUO operation (D8). The average proportions at each time point are plotted. **e** MuSiC estimated cell type proportions of rat renal tubule segments. The estimated cell type proportions (left) and the proportions correlations between samples (right) are shown as heatmap. Segment names are color coded and aligned according to their physical positions along the renal tubule. Supplementary Figure 6a–c show NNLS, BSEQ-sc and CIBERSORT results. Segment name abbreviation: S1 S1 proximal tubule, S2 S2 proximal tubule, S3 S3 proximal tubule, SDL short descending limb, LDLOM long descending limb, outer medulla, LDLIM long descending limb, inner medulla, tAL thin ascending limb, mTAL medullary thick ascending limb, cTAL cortical thick ascending limb, DCT distal convoluted tubule, CNT connecting tubule, CCD cortical collecting duct, OMCD outer medullary collecting duct, IMCD inner medullar collecting duct. Source data are provided as a Source Data file

## Methods

**MuSiC model set-up**. In this section, we derive the relationship between gene expression in bulk tissue and cell type-specific gene expression in single cells. This relationship forms the basis of our deconvolution procedure. For gene $g$, let $X_{jg}$ be the total number of mRNA molecules in subject $j$ of the given tissue, which is composed of $K$ cell types. Then, $X_{jg} = \sum_{k=1}^{K} \sum_{c \in C_j^k} X_{jgc}$, where $X_{jgc}$ is the number of mRNA molecules of gene $g$ in cell $c$ of subject $j$, and $C_j^k$ is the set of cell index for cell type $k$ in subject $j$ with $m_j^k = |C_j^k|$ being the total number of cells in this set. The

relative abundance of gene $g$ in subject $j$ for cell type $k$ is

$$\theta_{jg}^k = \frac{\sum_{c \in C_j^k} X_{jgc}}{\sum_{c \in C_j^k} \sum_{g'=1}^{G} X_{jg'c}}. \tag{1}$$

We can show that

$$X_{jg} = \sum_{k=1}^{K} m_j^k S_j^k \theta_{jg}^k = m_j \sum_{k=1}^{K} p_j^k S_j^k \theta_{jg}^k, \qquad (2)$$

where, for subject $j$, $S_j^k = \frac{\sum_{c \in C_j^k} \sum_{g'=1}^{G} X_{jg'c}}{m_j^k}$ is the average number of total mRNA molecules for cells of cell type $k$ (also referred to as "cell size" below), $m_j = \sum_{k=1}^{K} m_j^k$ is the total number of cells in the bulk tissue, and $p_j^k = \frac{m_j^k}{m_j}$ is the proportion of cells from cell type $k$. Let $Y_{jg} = \frac{X_{jg}}{\sum_{g'=1}^{G} X_{jg'}}$ be the relative abundance of gene $g$ in the bulk tissue of subject $j$. Equation (2) implies

$$Y_{jg} \propto \sum_{k=1}^{K} p_j^k S_j^k \theta_{jg}^k. \qquad (3)$$

Thus, across $G$ genes in subject $j$, we have

$$\begin{bmatrix} Y_{j1} \\ \vdots \\ Y_{jG} \end{bmatrix} \propto \begin{bmatrix} \theta_{j1}^1 & \cdots & \theta_{j1}^K \\ \vdots & \ddots & \vdots \\ \theta_{jG}^1 & \cdots & \theta_{jG}^K \end{bmatrix} \cdot \begin{bmatrix} S_j^1 & & \\ & \ddots & \\ & & S_j^K \end{bmatrix} \cdot \begin{bmatrix} p_j^1 \\ \vdots \\ p_j^K \end{bmatrix}. \qquad (4)$$

The goal of MuSiC is to estimate $p_j^k$ using data from scRNA-seq and bulk RNA-seq.

**Model assumptions**. If scRNA-seq data were available for subject $j$, we would be able to obtain the cell size factor $S_j^k$ (or the relative values of $S_j^k$, see below) and cell type-specific relative abundance $\theta_{jg}^k$. With bulk RNA-seq data in subject $j$, we get the bulk tissue relative abundance $Y_{jg}$, and, if $\theta_{jg}^k$ and $S_j^k$ were known, we would be able to perform a regression to estimate $p_j^k$. However, since scRNA-seq is still costly, most studies cannot afford the sequencing of a large number of individuals using scRNA-seq. To make deconvolution possible for a broader range of studies, it is desirable to utilize cell type-specific gene expression from other studies or from a smaller set of individuals in the same study. This is feasible under the following three assumptions.

(A1) Individuals with scRNA-seq and bulk RNA-seq are from the same population, with their cell-type specific relative abundances $\theta_{jg}^k$ in Equation (1) following the same distribution with mean $\theta_g^k$ and variance $\sigma_{gk}^2$,

$$\theta_{jg}^k \sim F\left(\theta_g^k, \sigma_{gk}^2\right). \qquad (5)$$

Here, $F(\cdot, \cdot)$ represents a general distributional function, which is not assumed to be of any particular form. Under this assumption, deconvolution can use available single-cell data from other subjects or even subjects from other studies as reference.

(A2) The ratio of cell size $S_j^k$ across cell types are the same across subjects and studies:

$$\frac{S_j^k}{S_j^{k'}} = \frac{S_{j'}^k}{S_{j'}^{k'}} \quad \text{for all subjects } j, j' \in \{1, \dots, N\} \text{ and cell types } k, k' \in \{1, \dots, K\}. \qquad (6)$$

This second assumption allows us to replace $S_j^k$ by a common value $S^k$ across subjects. We want to emphasize that we assume the ratio, and not the absolute value, of cell size to be constant across subjects and studies, because to utilize the common value $S^k$, we need a constant scalar in Equation (8) as shown below.

In practice, we do not observe the actual cell sizes $S_j^k$, since (1) for non-UMI data we observe read counts, not molecule counts and (2) for each cell we observe library size, not cell size. Let $\widetilde{X}_{jg}$ and $\widetilde{X}_{jgc}$ denote the read counts for a bulk sample and for a specific cell $c$ in the sample, respectively. Let $\widetilde{S}_j^k = \frac{\sum_{c \in C_j^k} \sum_{g'=1}^{G} \widetilde{X}_{jg'c}}{m_j^k}$ denote the average library size of cell type $k$ for subject $j$. We define the efficiency of cell type $k$ for subject $j$ as $\gamma_j^k = \widetilde{S}_j^k / S_j^k$. We assume

(A3) The ratio of average library size is the same across cell types regardless of subjects and studies

$$\frac{\widetilde{S}_j^k}{\widetilde{S}_j^{k'}} = \frac{\widetilde{S}_{j'}^k}{\widetilde{S}_{j'}^{k'}} \quad \text{for all } j, j' \in \{1, \dots, N\} \text{ and } k, k' \in \{1, \dots, K\}. \qquad (7)$$

Combined with assumption (A2), Equation (7) is equivalent to assuming that the ratio of efficiency between cell types is conserved across subjects and studies

$$\frac{\gamma_j^k}{\gamma_j^{k'}} = \frac{\gamma_{j'}^k}{\gamma_{j'}^{k'}} \quad \text{for all } j, j' \in \{1, \dots, N\} \text{ and } k, k' \in \{1, \dots, K\}.$$

This assumption seems plausible, since although efficiency varies across cell types and samples, its ratio between cell types should be less variable. Assumptions (A2) and (A3) allow us to use the common value of library size $\widetilde{S}^k$ across subjects in the read counts setting. (A1–A3) enable us to recover the trend of cell type proportion

change across subjects, as shown in Results, but does not enable the recovery of absolute cell type proportions.

To recover absolute cell type proportions, a stronger version of (A3) is needed, which we call (A3'): The ratio of average library size is equal to the ratio of average cell size, for all pairs of cell types and across all subjects and studies

$$\frac{\widetilde{S}_j^k}{\widetilde{S}_j^{k'}} = \frac{S_j^k}{S_j^{k'}} = \frac{S_{j'}^k}{S_{j'}^{k'}} = \frac{\widetilde{S}_{j'}^k}{\widetilde{S}_{j'}^{k'}} \quad \text{for all } j, j' \in \{1, \dots, N\} \text{ and } k, k' \in \{1, \dots, K\}.$$

Given (A2), (A3') is equivalent to assuming that the efficiency $\gamma_j^k$ is the same across cell types, subjects and studies

$$\gamma_j^k = \gamma_{j'}^{k'} \quad \text{for all } j, j' \in \{1, \dots, N\} \text{ and } k, k' \in \{1, \dots, K\}.$$

This stronger assumption indicates that we can safely interchange the ratio of library size with the ratio of cell size to estimate cell type proportions. When this assumption is not satisfied, we can estimate the fraction of RNA molecules from each cell type, represented by $p_j^k \times S_j^k$, but the estimate of cell type proportion, $p_j^k$, will be biased.

**Cell type proportion estimation**. To estimate cell type proportions $\boldsymbol{p}_j = \{p_j^k, k = 1, \dots, K\}$, we need to consider two constraints: (C1) Non-negativity: $p_j^k \geq 0$ for all $j, k$; (C2) Sum-to-one: $\sum_{k=1}^{K} p_j^k = 1$ for all $j$. Because the bulk tissue and single-cell relationship derived in Equation (5) is a "proportional to" relationship, to satisfy the (C2) constraint, we need a normalizing constant $C_j$ so that

$$Y_{jg} = C_j \cdot \left( \sum_{k=1}^{K} p_{jk} S_k \theta_{jg}^k + \epsilon_{jg} \right), \qquad (8)$$

where $\epsilon_{jg} \sim N(0, \delta_{jg}^2)$ represents bulk tissue RNA-seq gene expression measurement noise. When cell type proportions $\boldsymbol{p}_j = \{p_j^k, k = 1, \dots, K\}$ and subject-specific relative abundances $\boldsymbol{\theta}_{jg} = \{\theta_{jg}^k, k = 1, \dots, K\}$ are known, the variance of bulk tissue gene expression measurement is

$$\text{Var}\left[ Y_{jg} | \boldsymbol{p}_j, \boldsymbol{\theta}_{jg} \right] = C_j^2 \delta_{jg}^2. \qquad (9)$$

Given only cell type proportions, the variance is

$$\begin{aligned} \text{Var}\left[ Y_{jg} | \boldsymbol{p}_j \right] &= E\left[ \text{Var}\left[ Y_{jg} | \boldsymbol{p}_j, \boldsymbol{\theta}_{jg} \right] \right] + \text{Var}\left[ E\left[ Y_{jg} | \boldsymbol{p}_j, \boldsymbol{\theta}_{jg} \right] \right] \\ &= C_j^2 \delta_{jg}^2 + \text{Var}\left[ C_j \cdot \sum_{k=1}^{K} p_{jk} S_k \theta_{jg}^k \right] \\ &= C_j^2 \delta_{jg}^2 + C_j^2 \cdot \sum_{k=1}^{K} p_{jk}^2 S_k^2 \text{Var}\left[ \theta_{jg}^k \right] = C_j^2 \delta_{jg}^2 + C_j^2 \sum_{k=1}^{K} p_{jk}^2 S_k^2 \sigma_{gk}^2 \\ &= 1/w_{jg} \end{aligned} \qquad (10)$$

Because of the heteroscedasticity of gene expression over genes, including the weight $w_{jg}$ can improve estimates. Since $\delta_{jg}^2$ is unknown, we will estimate the weight $w_{jg}$ iteratively, initialized by NNLS. MuSiC is robust and converges to the same value even with different starting points (Supplementary Note 6, Supplementary Figure 9).

Given that bulk and single-cell expression data are generated via different protocols, it may also be necessary to consider gene-specific protocol bias. We note that the difference between the grand average of the single-cell and bulk expression profiles does not necessarily reflect bias between protocols, because the difference between cell type proportions of single-cell and bulk expression data can also lead to expression differences of marker genes even in the absence of protocol bias. To address potential protocol bias between bulk and single-cell expression data, we add a gene- and subject-specific intercept in Equation (8), that is $Y_{jg} = C_j \cdot \left( \alpha_{jg} + \sum_{k=1}^{K} p_{jk} S_k \theta_{jg}^k + \epsilon_{jg} \right)$. After adjusting for the protocol bias, MuSiC can detect significant biological signals across protocols (Supplementary Figure 7, Supplementary Table 2).

MuSiC is a weighted non-negative least squares regression (W-NNLS), which does not require pre-selected marker genes. Indeed, the iterative estimation procedure automatically imposes more weight on informative genes and less weight on non-informative genes. Because it is a linear regression-based method, genes showing less cross-cell type variations will have low leverage, thus having less influence on the regression, whereas the most influential genes are those with high weight and high leverage. To illustrate this point, we also performed benchmarking experiments to show that applying MuSiC using all genes gives more accurate results than applying MuSiC using pre-selected marker genes, thus demonstrating that MuSiC's weighting scheme makes marker gene pre-selection unnecessary (Supplementary Figure 1c, Supplementary Figure 2). MuSiC can also deal with batch effect with its weighting scheme. When batch effect is present, the variance of relative abundance will generally increase for all cell types. This means that the batch effect with be absorbed in $\sigma_{kg}$, meaning that MuSiC not only up-weighs cross-subject consistent genes, but also cross-batch consistent genes. Thus, by down-

weighting cross-batch variable genes, MuSiC effectively deals with batch effects.

The weighting scheme in MuSiC enables automatic selection of marker genes for deconvolution, as supported by our findings from the pancreas and kidney data (marker genes are highlighted with colors in Supplementary Tables 5–7). However, we note that some of the top-ranked genes are not necessarily marker genes. This is because genes in MuSiC are weighed by the combined effect of cross-subject variation and cross-cell-type variation, which are very different concepts. The cross-subject variation measures the consistency of genes across subjects while the cross-cell-type variation measures the cell type specificity of genes. The top ranked non-markers genes for the analyses in Results tend to be consistently expressed across subjects, and are usually highly expressed. Although they are not exclusively expressed in a particular cell type, they are differentially expressed across cell types, thus offering power to differentiate different cell types. We believe that MuSiC benefit from these genes and hence yield more accurate cell type proportions than methods that only use marker genes in deconvolution.

**Recursive tree-guided deconvolution for closely related cell types**. Complex solid tissues often include closely related cell types with similar gene expression levels. Correlation in gene expression can lead to collinearity, making it difficult to reliably estimate cell type proportions, especially for less frequent and rare cell types. Although the collinearity problem can be improved by selecting marker genes through support vector regression, as is done in CIBERSORT[3] and BSEQ-sc[4], these approaches still have limited power to resolve similar cell types. In MuSiC, we introduce a recursive tree-guided deconvolution procedure based on a cell type similarity tree, which can be easily obtained through hierarchical clustering. In stage 1 of this procedure, cell types in the design matrix are divided into high-level clusters by hierarchical clustering with closely related cell types clustered together. Proportion for these cell type clusters are estimated using genes with small intra-cluster variance (cluster-consistent genes) using the above described W-NNLS. In stage 2, for cell types in each cluster, the cell type proportions are estimated using W-NNLS with genes displaying small intra-cell type variance, subject to the constraint on the pre-estimated cluster proportions. If necessary, more than two stages of recursion can be applied, with each stage separating the cell types within each large cluster into finer clusters, and using cluster-consistent genes to do W-NNLS subject to the constraint that fixes higher-level cluster proportions.

To illustrate this recursive tree-guided deconvolution procedure, we start with a simple case with four cell types and $G$ genes. Let $X_1, X_2, X_3, X_4$ represent cell type-specific expression in the design matrix, obtained from scRNA-seq, and let $Y$ be the gene expression vector in the bulk RNA-seq data. The relationship of bulk and single-cell data can be written as

$$\begin{pmatrix} Y^{(1)} \\ Y^{(2)} \end{pmatrix} = \begin{pmatrix} X_1^{(1)} & X_2^{(1)} & X_3^{(1)} & X_4^{(1)} \\ X_1^{(2)} & X_2^{(2)} & X_3^{(2)} & X_4^{(2)} \end{pmatrix} \begin{pmatrix} p_1 \\ p_2 \\ p_3 \\ p_4 \end{pmatrix} + \begin{pmatrix} \epsilon^{(1)} \\ \epsilon^{(2)} \end{pmatrix} \qquad (11)$$

where the superscripts (1) and (2) indicate two sets of genes. Suppose the four cell types are grouped into two clusters, $(X_1, X_2)$ and $(X_3, X_4)$. The first set of genes are those showing small intra-cluster variance in gene expression, that is, $X_1^{(1)} \approx X_2^{(1)}$ and $X_3^{(1)} \approx X_4^{(1)}$, whereas the second set of genes are the remaining genes.

*Stage 1*: Estimate cluster proportions $\pi_1 = p_1 + p_2$ and $\pi_2 = p_3 + p_4$,
$$Y^{(1)} = X_1^{(1)} \pi_1 + X_3^{(1)} \pi_2 + \epsilon^{(1)}. \qquad (12)$$

The cluster proportions, $\hat{\pi}_1$ and $\hat{\pi}_2$, are estimated by W-NNLS using intra-cluster homogenous genes.

*Stage 2*: Estimate cell type proportions $(p_1, p_2, p_3, p_4)$,
$$Y^{(2)} = X_1^{(2)} p_1 + X_2^{(2)} p_2 + X_3^{(2)} p_3 + X_4^{(2)} p_4 + \epsilon^{(2)}. \qquad (13)$$

The cell type proportions are estimated by W-NNLS using the remaining genes subject to the constraint that
$$\hat{p}_1 + \hat{p}_2 = \hat{\pi}_1, \text{ and } \hat{p}_3 + \hat{p}_4 = \hat{\pi}_2. \qquad (14)$$

**Interconversion of different gene expression measures**. MuSiC links bulk and single-cell gene expression by mRNA molecule counts. There are many measures of mRNA abundance, such as read counts, UMI counts, RPKM and TPM. As molecule counts are not observed in real studies, we approximate the molecule counts by read counts and estimate cell type proportions based on assumptions A1–A3. The interconversion between other gene expression measures and read count determines if MuSiC can utilize other measures as the input for deconvolution. One step in MuSiC estimation is the use of average library size as a proportional measure of average cell size for a given cell type, which is absent in normalized measurements of mRNA abundance such as RPKM and TPM.

For RPKM, we would need the average library size for each cell type to be provided, or the average cell size for each cell type to be obtained from other sources. Cell type proportions cannot be estimated by MuSiC with TPM information alone. Below, we derive the relationships of various types of gene expression measures in detail.

Let $L_g$ denote the length of gene $g$, and the corresponding RPKMs of bulk and single-cell data are denoted by $\widehat{X}_{jg}$ and $\widehat{X}_{jgc}$, respectively. For simplicity, we omit the $10^3$ scalar for now. By definition,

$$\widehat{X}_{jg} = \frac{\widetilde{X}_{jg}/L_g}{\sum_{g'=1}^{G} \widetilde{X}_{jg'}}, \quad \widehat{X}_{jgc} = \frac{\widetilde{X}_{jgc}/L_g}{\sum_{g'=1}^{G} \widetilde{X}_{jg'c}}, \qquad (15)$$

where $\widetilde{X}_{jg}$ and $\widetilde{X}_{jgc}$ denote the bulk and single cell read counts, respectively.

Based on the model set-up described earlier, we can show that the relationship between bulk and single-cell RPKMs is

$$\widehat{X}_{jg} \propto \frac{\widetilde{X}_{jg}}{L_g} = \sum_{k=1}^{K} \sum_{c \in C_j^k} \left( \frac{\widetilde{X}_{jgc}/L_g}{\sum_{g'=1}^{G} \widetilde{X}_{jg'c}} \cdot \sum_{g'=1}^{G} \widetilde{X}_{jg'c} \right) = \sum_{k=1}^{K} \sum_{c \in C_j^k} \widehat{X}_{jgc} \widetilde{S}_{jc} \qquad (16)$$

where $\widetilde{S}_{jc}$ is the library size of cell $c$. Equation (16) can be further approximated by

$$\widehat{X}_{jg} \propto \sum_{k=1}^{K} \sum_{c \in C_j^k} \widehat{X}_{jgc} \widetilde{S}_{jc} \approx \sum_{k=1}^{K} m_j^k \widehat{\theta}_{jg}^k \widetilde{S}_j^k = m_j \sum_{k=1}^{K} p_j^k \widehat{\theta}_{jg}^k \widetilde{S}_j^k \qquad (17)$$

where $\widehat{\theta}_{jg}^k = \sum_{c \in C_j^k} \widehat{X}_{jgc}/m_j^k$ is the average RPKM of gene $g$ in subject $j$ for cell type $k$.

To utilize multi-subject information, we assume $\widehat{\theta}_{jg}^k$ follows the same assumption as (A1), that is, individuals with scRNA-seq and bulk RNA-seq are from the same population, with their cell-type specific average RPKM $\widehat{\theta}_{jg}^k$ following the same distribution with mean $\widehat{\theta}_g^k$ and variance $\widehat{\sigma}_{gk}^2$,

$$\widehat{\theta}_{jg}^k \sim \tilde{F}\left(\widehat{\theta}_g^k, \widehat{\sigma}_{jg}^2\right). \qquad (18)$$

Assumption (A2) states that the ratio of average library size is consistent across subjects and studies, which justifies the use of $\widetilde{S}_j^k$ from other studies if these quantities are not available for the same data set. The linear relation between bulk RPKM and average cell-type specific single cell RPKM is approximated by formula (17). Since this is an approximation, MuSiC estimates using RPKM may not be as accurate as those using read or UMI count. In our test of MuSiC using RPKM values for the pancreatic islets bulk mixture experiment, we found that it is not as accurate as MuSiC estimates using read count, but still higher than NNLS, BSEQ-sc, and Cibersort (Supplementary Figure 8d).

Another widely used normalized mRNA measure is TPM. Let $\widehat{Z}_{jg}$ and $\widehat{Z}_{jgc}$ denote the bulk and single-cells TPM values, respectively. By definition, $\widehat{Z}_{jg} = \frac{\widetilde{X}_{jg}/L_g}{\sum_{g'=1}^{G} \widetilde{X}_{jg'}/L_{g'}}, \widehat{Z}_{jgc} = \frac{\widetilde{X}_{jgc}/L_g}{\sum_{g'=1}^{G} \widetilde{X}_{jg'c}/L_{g'}}$. Let $Z_{jg}$ and $Z_{jgc}$ be the gene length normalized read count in bulk and single cell, that is, $Z_{jg} = \widetilde{X}_{jg}/L_g$ and $Z_{jgc} = \widetilde{X}_{jgc}/L_g$. The link between bulk and single-cell TPMs is

$$\widehat{Z}_{jg} \propto Z_{jg} = \sum_{k=1}^{K} \sum_{c \in C_j^k} Z_{jgc} = \sum_{k=1}^{K} \sum_{c \in C_j^k} \left( \frac{Z_{jgc}}{\sum_{g'=1}^{G} Z_{jg'c}} \cdot \sum_{g'=1}^{G} Z_{jg'c} \right) = \sum_{k=1}^{K} \sum_{c \in C_j^k} \widehat{Z}_{jgc} \widehat{S}_{jc}, \qquad (19)$$

where $\widehat{S}_{jc}$ is the summation of normalized read counts in cell $c$ for subject $j$.

Equation (19) suggests that it is difficult to make assumptions or approximations to express relative abundance as a function of TPM.

**Construction of benchmark datasets and evaluation metrics**. To evaluate MuSiC and compare with other deconvolution methods, we need bulk RNA-seq data with known cell type proportions. Therefore, we construct artificial bulk tissue data from a scRNA-seq dataset in which the bulk data is obtained by summing up gene counts from all cells in the same subject. Relative abundance is calculated by Equation (1). The true cell type proportions in the artificial bulk data can be directly obtained from the scRNA-seq data and this allows us to use this artificially constructed bulk data as a benchmark dataset to evaluate the performance of different deconvolution methods (Supplementary Note 1). Denote the true cell type proportions by $\boldsymbol{p}$ and the estimated proportions by $\widehat{\boldsymbol{p}}$. Deconvolution methods are evaluated by the following metrics.

(i) Pearson correlation, $\text{R} = \text{Cor}(\boldsymbol{p}, \widehat{\boldsymbol{p}})$;
(ii) Root mean squared deviation, $\text{RMSD} = \sqrt{\text{avg}(\boldsymbol{p} - \widehat{\boldsymbol{p}})^2}$;
(iii) Mean absolute deviation, $\text{mAD} = \text{avg}(|\boldsymbol{p} - \widehat{\boldsymbol{p}}|)$.

**Reporting summary**. Further information on experimental design is available in the Nature Research Reporting Summary linked to this article.

**Code availability**. MuSiC is available on Github (https://github.com/xuranw/MuSiC). Tutorial and examples are provided.

## Data availability

This study was a re-analysis of existing data, which is openly available at locations cited in the Reference section. The detailed data summary are provided in Tables 1 and 2. The source data underlying Figs. 2a–d, 3b–e and Supplementary Figure 1c, Supplementary Figures 2–8 are provided as a Source Data file. A reporting summary for this article is available as a Supplementary Information file. All relevant data is available upon request.

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

## Acknowledgements

This work was supported by the following funding: NIH R01HG006137 (to N.R.Z.); NIH R01GM125301 (to N.R.Z., M.L.); NIH R01GM108600 (to M.L.); NIH R01HL113147 (to M.L.); NIH R01DK076077 (to K.S., M.L.); NIH R01DK087635 (to K.S.); NIH R01DK105821 (to K.S.); ADA postdoctoral fellowship (to J.P.).

## Author contributions

This study was conceived of and led by N.R.Z. and M.L. Jointly with N.R.Z. and M.L., X.W. designed the model and estimation algorithm, implemented the MuSiC software, designed the in silico experiments, and led the data analysis. J.P. and K.S. performed the mouse scRNA-seq experiment and provided scientific insight on chronic kidney disease and data interpretation. X.W., N.R.Z., and M.L. wrote the paper with feedback from J.P. and K.S.

## Additional information

**Competing interests:** The authors declare no competing interests.

