## [Peer Review File · Nature Communications]

Reviewers' comments:

Reviewer #1 (Remarks to the Author):

Wang et al. present a new tool "MuSIC" for determining cell type compositions in bulk RNA-seq data based on reference single-cell expression profiles. Their method outperforms a number of existing approaches on simulated data sets, and recovers some known biological features in real data. The manuscript is clear, concise and addresses a relevant problem in data integration across different transcriptomic assays. The method is interesting and the use of the guide tree to resolve similar cell types is novel. However, I have a few concerns with the general applicability of the tools as well as some of the statistical justifications. I have provided more details in my comments below.

1. One obvious concern lies in the differences in the protocols between scRNA-seq and bulk RNA-seq. For example, the TruSeq protocol is commonly used for bulk RNA-seq, which provides full-length strand-specific transcript coverage. In comparison, other common protocols for scRNA-seq (e.g., CEL-seq2, MARS-seq, 10X Genomics) use unique molecular identifiers and do not provide full-length coverage. Even the Smart-seq2 protocol (as used in the Segerstople study) differs from TruSeq in that it is oligo-dT primed, not strand-specific, and involves a larger number of PCR cycles that can introduce a different set of amplification and length biases.

It is easy to see how the presence of different biases between protocols can pose problems with deconvolution. Consider two genes - one long, one short - that have the same number of molecules in a particular cell type. With TruSeq, the longer gene would have more coverage in the bulk expression profile (Y_{jg}). On the other hand, with UMI-based scRNA-seq protocols, the two genes would have the same number of UMIs (X_{jgc}). Trying to force the latter to fit the former will compromise the accuracy of the deconvolution.

Now, it may be the case that these protocol discrepancies are not a big factor in the performance of MuSIC. Certainly, the kidney scRNA-seq data set was generated using a droplet-based protocol with UMIs, and the one example in Figure 3 suggests that it still gives good performance in deconvolving bulk data sets. Nonetheless, it seems necessary to address this issue more explicitly. I would like to see a more comprehensive examination of the robustness to protocol differences, in addition to that offered in Figure 2b. Namely, how much bias can be tolerated (relative to the biological signal) before MuSIC fails?

I would further suggest that the authors consider an additional gene-specific term for protocol bias, to scale θ^k_{jg} in Equation 3 to model differences in the bulk and single-cell protocols. These parameters could conceivably be estimated by taking advantage of assumption A1 on page 9. If the scRNA-seq and bulk RNA-seq data are obtained from individuals of the same population (and assuming that no major cell populations are lost during dissociation), the scaling differences between the grand average of the single-cell and bulk expression profiles should provide a suitable estimate of the biases between protocols.

2. The unqualified use of the library size as an estimate of the RNA content (i.e., cell size) is naive. If cDNA quantification was performed prior to multiplexing, there will be little association between library size and RNA content. Similarly, droplet-based technologies often exhibit saturation whereby further increases in RNA content do not manifest as (linear) increases in the total UMI count. There are also complicating issues with differences in sequencing depth between subjects or batches in large data sets. All of these factors mean that, in many cases, the library size is a poor estimate of the RNA content.

Inaccurate estimation of the cell size will compromise the interpretation of the estimated proportions p^k_j . The proportions would become some intermediate between the percentage of cells of a particular type and the percentage of RNA originating from that cell type. Now, I do not think this is a critical problem, as the proportions can still be compared between conditions. Nonetheless, it should be discussed in some more detail in the manuscript. The authors may also consider using spike-in information to provide a more accurate estimate of the relative RNA content per cell, see the discussion in <https://doi.org/10.1101/gr.222877.117>.

3. I found the choice of distributions to be highly irregular. For example, θ^k_{jg} is assumed to follow a F-distribution in Equation 5. Notwithstanding the odd notation (the F-distribution is typically characterized in terms of its two d.f., not its mean and variance), the F-distribution is defined across the range of positive real numbers. However, θ^k_{jg} is bounded in $[0, 1]$, which is quite different from the F-distribution's support! Similarly, in Equation 8, the error is defined as being normally distributed, but Y_{jg} is defined as a proportion that should lie within $[0, 1]$. This is only justified if δ^2_{jg} is so small that the bounds on the proportion are irrelevant. The authors should show that their estimates of δ^2_{jg} satisfy this criterion.

4. It seems to me that W-NNLS is not guaranteed to converge to a global minimum. For example, consider two genes g_1 and g_2 with the same properties. I will assume that δ^2_{jg} (and thus the weight) is estimated from the error between Y_{jg} and the first term of the LHS of Equation 8 computed with the starting values of p_{jk} .

If the starting conditions yield a slightly smaller error for gene g_1 than g_2 , then (all else being equal) the weight for g_1 will be increased relative to that for g_2 . This means that g_1 will have a greater influence over the re-estimation of p_{jk} , and would naturally aim to minimize its error. Over multiple reweighting iterations, the final result will be fully driven by g_1 . However, with a different set of starting conditions, g_2 might end up being dominant, which would be a problem if the two genes support different values of p_{jk} !

The sensitivity of the results to the starting conditions is a common problem in feature reweighting approaches such as sparse clustering. An ad hoc solution might be to perform multiple runs with different starting values for the W-NNLS algorithm, to provide a measure of confidence in the proportion estimates. A more statistically rigorous approach would be to constrain the re-weighting. This could be achieved, for example, by estimating a single variance parameter for each gene; or taking the variance for each observation from a fitted mean-variance trend (see <https://doi.org/10.1186/gb-2014-15-2-r29>) based on the value of the LHS in Equation 8.

Reviewer #2 (Remarks to the Author):

This paper introduces a machine learning approach that is proposed as a means to identify the cellular make up of a mixed tissue sample. It uses examples from pancreatic islets and kidney. The method works better for islets because they are made up of a limited number of cell types. It seemingly does not take into account many cell types represented in kidney samples. The following are recommendations for improvement of the kidney part of the manuscript.

Recommendations:

A. For clarity, authors should describe limitations of method from a biological perspective:

1. requires pre-specification of cell types.
2. assumes that mRNA abundances for cell selective genes are not affected by physiological state or

pathophysiological state. (Presumably the index or signature transcripts are not measured in all possible alternative physiological or pathophysiological states.)

A1. --- With regard to the cell types assumed ($n=13$), this falls far below the number of cell types actually in the kidney. So there is a kind of transcriptomic 'dark matter' that is ignored in the calculations. It seems likely that these un-accounted for cells make up as much as a third of the total mRNA. Same problem exists with CIBERSORT and other algorithms.

A2. --- supplementary resource material should be included which enumerates the transcripts that the program identified for each cell types as discriminators. This actually could be the most valuable aspect of the paper if done and would certainly increase the likelihood that it will be cited.

B. It is not clear how well the different measures of mRNA abundance interconvert, e.g. TPM, RPKM, drop-seq counts.

C. There is a danger of overfitting when running the model. Authors should add additional sensitivity analysis to assure reader that the cell number estimates are not sensitive to small changes in assumptions.

D. scRNA-Seq data are highly variable for a given cell type and tend not to be normally distributed. Were mean values used. Would it come out the same if medians were used.

E. DCT is far from the second most abundant epithelial cell type in kidney. Thick ascending limb cells are at least three times more plentiful. Authors ignore their own warning that cell isolation for scRNA-Seq is biased toward certain cell type.

F. The authors' conclusion that DCT is similar to PCT does not measure up to knowledge about the structure and function of these two cell types. This is based on dichotomous clustering that does not really tell how close they are. K-means clustering should be better since cell types are prespecified. Also, for the reader it would be good to pull out the transcript values that are the basis of this classification.

G. The authors coin a term 'stable' to describe variability. An explicit definition is needed. This is probably a bad term because biologist that investigate mRNA abundance regulation use the term 'stability' to refer to mRNA half life.

H. The analysis of the rat data from Lee et al. describes the measurements as "bulk rat RNA-seq data". Although the Lee samples are multicellular, they are made up of a single cell type. So Lee's DCT samples were already 100 percent DCT cells. Authors should remove the word 'bulk' and explain that these are essentially heterogeneous samples. The interpretation is stated as, "knowledge about the dominant cell type at its mapped position, e.g. DCT cells come from the DCT region". 'Region' is the wrong word here. 'Segment' is more appropriate for microdissected tubules.

Minor. The GitHub link seems to be poorly organized. It needs a more extensive 'read-me' file that gives the reader a guide to the files available and how to use them.

Reviewer #3 (Remarks to the Author):

This manuscript presents a new statistical method and an open source R package call MuSiC, to identify and estimate the proportion of individual cell types in bulk RNA-seq sample, using multiple single-cell RNA-seq data sets as a reference. The key innovative ideas in this methodology include: (1) using cross-subject and cross-cell stability as a measure to identify good cell-type specific markers; and (2) a recursive tree-guided deconvolution scheme, which is helpful in discovering and estimating low-frequency cell types. The authors have shown the applicability of their method using two case studies - pancreatic islets in humans, and a cross-species analysis of kidney cells. The paper is well written, the method is logical and clearly presented, and the experiment results seem solid. I like the authors' idea of using scRNA-seq data as reference for deconvolution of bulk RNA-seq data. This paper clearly contains methodological innovation.

Nonetheless, to fully evaluate the real-life applicability of MuSiC, I have the following questions:

1. scRNA-seq data are known to contains a high proportion of signal dropouts and measurement variability. I suspect it is difficult to use multiple scRNA-seq data with different dropout rate as a reference. Do the authors have any experimental data (using simulation data, for example) to show that MuSiC is robust to different level of noise in the scRNA-seq data?
2. Have you considered the impact of normalisation and batch effect correction to the performance of MuSiC?
3. The detail for performing the cross-species analysis (end of page 7) was missing. Did the authors assume there is a 1-1 mapping of homologous genes between two species? How did they deal with complex homology relationships? I suspect the claim about 'cross-species applicability' probably only extend to mammals, but not other organisms that are further away in the evolutionary tree.

Response to Reviewer #1

Wang et al. present a new tool "MuSiC" for determining cell type compositions in bulk RNA-seq data based on reference single-cell expression profiles. Their method outperforms a number of existing approaches on simulated data sets, and recovers some known biological features in real data. The manuscript is clear, concise and addresses a relevant problem in data integration across different transcriptomic assays. The method is interesting and the use of the guide tree to resolve similar cell types is novel. However, I have a few concerns with the general applicability of the tools as well as some of the statistical justifications. I have provided more details in my comments below.

1. One obvious concern lies in the differences in the protocols between scRNA-seq and bulk RNA-seq. For example, the TruSeq protocol is commonly used for bulk RNA-seq, which provides full-length strand-specific transcript coverage. In comparison, other common protocols for scRNA-seq (e.g., CEL-seq2, MARS-seq, 10X Genomics) use unique molecular identifiers and do not provide full-length coverage. Even the Smart-seq2 protocol (as used in the Segerstolpe study) differs from TruSeq in that it is oligo-dT primed, not strand-specific, and involves a larger number of PCR cycles that can introduce a different set of amplification and length biases.

It is easy to see how the presence of different biases between protocols can pose problems with deconvolution. Consider two genes - one long, one short - that have the same number of molecules in a particular cell type. With TruSeq, the longer gene would have more coverage in the bulk expression profile (Y_{jg}). On the other hand, with UMI-based scRNA-seq protocols, the two genes would have the same number of UMIs (X_{jgc}). Trying to force the latter to fit the former will compromise the accuracy of the deconvolution.

Now, it may be the case that these protocol discrepancies are not a big factor in the performance of MuSiC. Certainly, the kidney scRNA-seq data set was generated using a droplet-based protocol with UMIs, and the one example in Figure 3 suggests that it still gives good performance in deconvolving bulk data sets. Nonetheless, it seems necessary to address this issue more explicitly. I would like to see a more comprehensive examination of the robustness to protocol differences, in addition to that offered in Figure 2b. Namely, how much bias can be tolerated (relative to the biological signal) before MuSiC fails?

I would further suggest that the authors consider an additional gene-specific term for protocol bias, to scale θ_{jg}^k in Equation 3 to model differences in the bulk and single-cell protocols. These parameters could conceivably be estimated by taking advantage of assumption A1 on page 9. If the scRNA-seq and bulk RNA-seq data are obtained from individuals of the same population (and assuming that no major cell populations are lost during dissociation), the scaling differences between the grand average of the single-cell and bulk expression profiles should provide a suitable estimate of the biases between protocols.

Response: Thanks for raising these questions, which are indeed very important for the general applicability of this method. Based on our reading, we believe that you asked 3 questions:

- (a) What is the estimation accuracy when using cross-protocol single-cell reference?
- (b) How much bias can be tolerated (relative biological signal) by MuSiC?
- (c) Should we add an additional gene-specific term for protocol bias to scale θ_{jg}^k so that differences in bulk and single-cell protocols can be modeled?

Below we address each of these three questions in turn:

- (a) We want to point out that the results in **Figures 2 and 3** in the maintext are obtained using different protocols. The Segerstolpe et al. and Xin et al. data, in Figure 2, used Smart-seq2, and the Park et al. data in Figure 3 used 10X. Furthermore, we used the 10X data from Park et al. to deconvolve three bulk RNA-seq data sets from different labs (**Figure 3, maintext**). To further examine the impact of cross-protocol single-cell reference on the performance of MuSiC, we added one more single cell dataset from Baron et al. (GSE84133), which sequenced pancreas islet from 3 healthy subjects via InDrop, an UMI-based approach. Using Baron et al. data as reference, we redid the deconvolution analysis of the bulk data from Fadista et al. Those results have been added

as **Supplementary Figure 7a and b in the maintext**. The new deconvolution results are also shown below in Figures 1, 2 and Table 1.

Figure 1: Jitter plots of estimated proportions of the 6 major cell types in pancreatic islet. Bulk data is from Fadista et al. and the UMI-based single cell data from Baron et al. is used to derive the reference. Results here are consistent with the deconvolution of this same bulk data set using the data from Xin et al. and Segerstolpe et al., which use the non-UMI Smart-Seq technology as the single-cell reference. Specifically, MuSiC recovered more reasonable beta cell proportions that decrease with HbA1c, while the other methods severely under estimate the beta cell proportions and miss this known negative relationship.

Figure 2: HbA1c level v.s. estimated beta cell proportions by MuSiC, NNLS, BSEQ-sc and CIBERSORT. P-values of single variable linear regression of beta cell proportions $\sim 1 + \text{HbA1c}$ are also shown in this figure. The coefficients and p-values of multivariable regression are shown in Table 1. Consistent with our results when using Segerstolpe and Xin et al. as single-cell reference, we recovered the expected negative relationship between HbA1c level and beta cell proportions.

Method	Coefficients	P-value
MuSiC	-2.84×10^{-2}	0.027
NNLS	-4.71×10^{-3}	0.111
BSEQ-sc	-3.70×10^{-5}	0.987
CIBERSORT	-5.72×10^{-3}	0.279

Table 1: Coefficients and p-value of HbA1c level to estimated beta cell proportion, adjusted by age, bmi and gender.

Remark:

Baron et al. provides single cell data via InDrop in only 3 healthy subjects. Due to the limited sample size, cross-subject variances are not reliably estimated. However, it is encouraging to see that even with such inaccurate estimates of cross-subject variance, MuSiC still detects a significant negative association of beta cell proportion with HbA1c level (p-value 0.027 after adjusting for age, bmi and gender.), improving upon the results of existing approaches.

We also want to point out that, in the Fadista-Baron analysis, we added a gene-specific and subject-specific intercept account for protocol bias in MuSiC estimation (for details please see Response 1(c)).

We describe the new Fadista-Baron analysis in the **revised maintext** on page 4, line 159-162 and added **Supplementary Figure 7** and **Supplementary Table 2** in the Supplementary Information.

- (b) To investigate the effect of bias between protocols on MuSiC, we added bias to the single cell reference θ_{jg}^k (the cell type specific relative abundance values) for data analyzed in **Figure 2b in the maintext**.

Technically, for relative abundance, there is a constraint $\sum_{g=1}^G \theta_{jg}^k = 1$. Due to this constraint, we choose to use Dirichlet distribution to add bias, and the biased version of the relative abundance is denoted as $\theta_{jg}^{k'}$. For a fixed cell type and subject (drop subscript j and superscript k for simplicity)

$$(\theta'_1, \dots, \theta'_G) \sim \text{Dirichlet}(t \times (\theta_1, \dots, \theta_G)), \tag{1}$$

where t is a scaling factor. The mean and variance of θ'_g are

$$E[\theta'_g] = \theta_g, \quad \text{Var}[\theta'_g] = \frac{\theta_g(1 - \theta_g)}{t + 1}.$$

We set $t = 999, 1332, 1999$ and 3999 (corresponding to $\text{Var}[\theta'_g]/E^2[\theta'_g] \approx (\theta_g(1 + t))^{-1} \geq 2, 1.5, 1$ and 0.5) with decreased variance of θ_g . We simulated 100 sets of biased θ_{jg}^k and computed the RMSD, mAD, and correlation (R) of the estimated cell type proportions with real values. These results are shown in Figure 3. Larger value of R and smaller values of RMSD and mAD indicate higher accuracy.

Figure 3: Evaluation of estimation accuracy of MuSiC by mAD, RMSE and R in the presence of bias in the relative abundance vectors derived from single cell reference. Larger values of the scale parameter reflect smaller total bias, which should lead to higher estimation accuracy. We wish to point out that, compared to the existing methods of NNLS, BSEQ-sc and CIBERSORT, MuSiC estimation with biased relative abundance is still much more accurate: for example, the R value of BSEQ-sc is 0.788, but when bias $t = 999$, the median R value of MuSiC is 0.8514.

We have included this robustness analysis in the revised maintext on page 6, line 246-249. We also added **Supplementary Figure 8c** and **Supplementary Note 4** in the Supplementary Information.

- (c) Thanks for your helpful suggestion on adding gene-specific protocol bias, which is a good idea. We have carefully considered your suggested approach, however, we feel that it may not be appropriate due to the following reason. The reviewer’s suggested approach is based on the assumption that ‘*If the scRNA-seq and bulk RNA-seq data are obtained from individuals of the same population, the scaling differences between the grand average of the single-cell and bulk expression profiles and difference can exist should provide a suitable estimate of the biases between protocol*’. However, we wish to point out that the protocol biases, the difference between cell type proportions can also lead to difference in addition to grand average of the single-cell and bulk expression profiles, and such difference can exist even if there is no protocol biases. For example, suppose the majority of cell type A is lost during scRNA-seq process. The bulk expression of cell type A marker genes would differ tremendously from the grand average of the single-cell expression, but this difference has nothing to do with protocol biases. Due to this reason, we feel that the approach suggested by the reviewer may not be appropriate.

Instead, we found a more direct way of adjusting for gene-specific protocol bias by simply adding an intercept term α_{jg} in **Equation 8** in the revised maintext.

$$Y_{jg} = C_j(\alpha_{jg} + \sum_{k=1}^K p_{jk} S_k \theta_{gk} + \epsilon_{jg}). \quad (2)$$

By adding the intercept term α_{jg} , MuSiC can accurately estimate cell type proportions when protocol biases are present. The new results using the Baron et al. reference to deconvolve the Fadista et al. pancreatic islets data (see Response 1(a)) is an example of how adding α_{jg} effectively removes bias. Baron et al. is a single cell dataset

via InDrop while Fadista et al. is a bulk dataset, in which the protocol is very different from InDrop. Without adding the intercept term α_{jg} , we can not detect the association between HbA1c level and beta cell proportions. After adjusting protocol bias by adding α_{jg} , we detected the negative correlation between beta cell proportions and HbA1c levels (Figure 1 and 2, Table 1). Interestingly, note that in the deconvolution of the three bulk kidney data sets using an UMI-based single cell data (Park et al.) was pretty accurate without this intercept term.

We have included the Fadista-Baron analysis in the **revised maintext** on page 4, line 159-162 and added **Supplementary Figure 7** and **Supplementary Table 2** in the Supplementary information. We also added a discussion of the difference between protocols in **Method** section on page 10, line 384-393.

2. *The unqualified use of the library size as an estimate of the RNA content (i.e., cell size) is naive. If cDNA quantification was performed prior to multiplexing, there will be little association between library size and RNA content. Similarly, droplet-based technologies often exhibit saturation whereby further increases in RNA content do not manifest as (linear) increases in the total UMI count. There are also complicating issues with differences in sequencing depth between subjects or batches in large data sets. All of these factors mean that, in many cases, the library size is a poor estimate of the RNA content.*

Inaccurate estimation of the cell size will compromise the interpretation of the estimated proportions p_j^k . The proportions would become some intermediate between the percentage of cells of a particular type and the percentage of RNA originating from that cell type. Now, I do not think this is a critical problem, as the proportions can still be compared between conditions. Nonetheless, it should be discussed in some more detail in the manuscript. The authors may also consider using spike-in information to provide a more accurate estimate of the relative RNA content per cell, see the discussion in <https://doi.org/10.1101/gr.222877.117>.

Response: Thanks for pointing out this ambiguity. Your concern is because we didn't explain Assumption (A2), and our use of library size, clearly: We don't require that cell size be estimated from the data, and we are not using library size to directly estimate cell size. Assumption (A2) states that the ratio of the average cell size S_k^j between pairs of cell types are constant across subjects

$$\frac{S_j^k}{S_j^{k'}} = \frac{S_{j'}^k}{S_{j'}^{k'}}, \quad \text{for all } j, j' \in \{1, \dots, N\} \text{ and } k, k' \in \{1, \dots, K\}.$$

This seems reasonable, as, for example, if cell type A is on average twice the size of cell type B in subject 1, it should also be on average twice the size of cell type B in subject 2. Importantly, this assumption only assumes that the ratio is held fixed, and thus, if we assume that cell size is equal to library size multiplied by an unknown cell-specific efficiency parameter, this assumption only requires that the average efficiency be constant across cell types. Furthermore, if our goal is only to recover the relative trends in cell type proportions across subjects in the bulk data, and not to estimate the absolute proportions (for example, to detect the negative association of beta cells with HbA1c instead of estimating absolute beta proportion), then all we need is for the ratio of average efficiencies between cell types to be constant across subjects. This is why we added a constant scalar C in the **revised maintext Equation 8**.

Of course using spike-in will increase estimation accuracy when spike-in counts are available. We incorporate the spike-in estimated library size ratio obtained from other studies to increase deconvolution accuracy.

We have added more explanations on assumption (A2) in the **revised maintext** on page 8, line 331-333. We have also added new assumptions (A3) and (A3') to explain the difference and interconversion between cell size and library size in the **revised maintext**, on page 8-9, lines 335-362.

3. *I found the choice of distributions to be highly irregular. For example, θ_{jg}^k is assumed to follow a F -distribution in Equation 5. Notwithstanding the odd notation (the F -distribution is typically characterized in terms of its two d.f., not its mean and variance), the F -distribution is defined across the range of positive real numbers. However, θ_{jg}^k is bounded in $[0, 1]$, which is quite different from the F -distribution's support! Similarly, in Equation 8, the error is defined as being normally distributed, but Y_{jg} is defined as a proportion that should lie within $[0, 1]$. This is only justified if δ_{jg}^2 is so small that the bounds on the proportion are irrelevant. The authors should show that their estimates of δ_{jg}^2 satisfy this criterion.*

Response: We apologize for the ambiguity in our notation. The ‘F’ in Equation 5 does not represent F-distribution, but is rather an arbitrary distribution with mean θ_{gk} and variance σ_{gk}^2 . We have clarified this in the **revised maintext Methods** section.

As for Equation 8, constant C is a scaling constant,

$$Y_{jg} = C \left(\sum_{k=1}^K p_{jk} S_k \theta_{gk} + \epsilon_{jg} \right), \quad (3)$$

where $\epsilon_{jg} \sim N(0, \delta_{jg}^2)$ represents bulk tissue RNA-seq gene expression measurement noise.

We have updated Equation (8-10) in the **revised maintext** on page 9-10.

4. *It seems to me that W-NNLS is not guaranteed to converge to a global minimum. For example, consider two genes g_1 and g_2 with the same properties. I will assume that δ_{jg}^2 (and thus the weight) is estimated from the error between Y_{jg} and the first term of the LHS of Equation 8 computed with the starting values of p_{jk} . If the starting conditions yield a slightly smaller error for gene g_1 than g_2 , then (all else being equal) the weight for g_1 will be increased relative to that for g_2 . This means that g_1 will have a greater influence over the re-estimation of p_{jk} , and would naturally aim to minimize its error. Over multiple reweighting iterations, the final result will be fully driven by g_1 . However, with a different set of starting conditions, g_2 might end up being dominant, which would be a problem if the two genes support different values of p_{jk} !*

The sensitivity of the results to the starting conditions is a common problem in feature re-weighting approaches such as sparse clustering. An ad hoc solution might be to perform multiple runs with different starting values for the W-NNLS algorithm, to provide a measure of confidence in the proportion estimates. A more statistically rigorous approach would be to constrain the re-weighting. This could be achieved, for example, by estimating a single variance parameter for each gene; or taking the variance for each observation from a fitted mean-variance trend (see <https://doi.org/10.1186/gb-2014-15-2-r29>) based on the value of the LHS in Equation 8.

Response: Thanks for taking note of the convergence issue, and we agree that it is worth checking.

To examine the convergence of W-NNLS, we re-analyzed the data in **Figure 2b in the maintext** to test convergence with different starting points, shown in Table 2 below:

celltype	EQ	Sp1	Sp2	Sp3	Sp4	Sp5	Sp6	Sp7	Sp8
alpha	0.25	0.4	0.2	0.2	0.2	0.7	0.1	0.1	0.1
beta	0.25	0.2	0.4	0.2	0.2	0.1	0.7	0.1	0.1
delta	0.25	0.2	0.2	0.4	0.2	0.1	0.1	0.7	0.1
gamma	0.25	0.2	0.2	0.2	0.4	0.1	0.1	0.1	0.7

Table 2: Note: EQ represents equal starting point of alpha, beta, delta and gamma are the same: (0.25, 0.25, 0.25, 0.25); Sp represents starting point.

The MuSiC estimates for the four cell types in this example are shown in Figure 4.

Figure 4: Estimated cell type proportions with different starting points. The colors represent cell types while line types represent starting point.

The plots show that W-NNLS converges to the same value regardless of the starting point.

We also considered your suggestion of adding a constraint on the mean-variance relation, which seemed like a good idea. You suggested *voom*, where genes with low mean expression have high variance. But we checked the empirical mean-variance relation by plotting fitted values versus residuals, shown in Figure 5. However, we didn't observe the expected mean-variance relationship of the *voom* paper. Given this, we are not clear how to use the *voom*-based approach to add a constraint in the estimation. Also, the plots seem to show that there is a pretty tight relationship between mean and variance, and so if we were to do some sort of shrinkage, or add a constraint, it wouldn't have a big effect. Due to these considerations, we decided to leave this direction for future investigation.

Figure 5: Scatter plot of log-scaled fitted relative abundance and log-scaled residuals. Bulk data is constructed by Xin et al. and the single cell reference is from Segerstolpe et al. Fitted value and residuals are estimated by MuSiC.

According to reviewer's suggestion, we added discussion on the convergence in the **Method** section on page 10, line 381-382. We also added **Supplementary Note 6** and **Supplementary Figure 8**.

Response to Reviewer #2

This paper introduces a machine learning approach that is proposed as a means to identify the cellular make up of a mixed tissue sample. It uses examples from pancreatic islets and kidney. The method works better for islets because they are made up of a limited number of cell types. It seemingly does not take into account many cell types represented in kidney samples. The following are recommendations for improvement of the kidney part of the manuscript.

1. *For clarity, authors should describe limitations of method from a biological perspective:*
 1. *requires pre-specification of cell types.*
 2. *assumes that mRNA abundances for cell selective genes are not affected by physiological state or pathophysiological state. (Presumably the index or signature transcripts are not measured in all possible alternative physiological or pathophysiological states.)*
- (a) *—With regard to the cell type assumed ($n = 13$), this falls far below the number of cell types actually in the kidney. So there is a kind of transcriptomic ‘dark matter’ that is ignored in the calculations. It seems likely that these un-accounted for cells make up as much as a third of the total mRNA. Same problem exists with CIBERSORT and other algorithms.*

Response: Thanks for raising this question. Yes, the cell type assumed is far below number of cell types actually in the kidney in two ways: (i). Not all cell types in the kidney can be detected via single cell technique. This part has been addressed by missing cell type analysis in **Supplementary Note 3** and **Supplementary Figure**

- 3.** (ii). Sub-cell types can not be identified and separated via provided single cell data. However, our method aims at the overall trend of proportions and has already achieve the goal of trend recover of immune cells. We can benefit higher resolution from the tree-based recursive estimation when sub-cell types are provided.
- (b) *supplementary resource material should be included which enumerates the transcripts that the program identified for each cell types as discriminators. This actually could be the most valuable aspect of the paper if done and would certainly increase the likelihood that it will be cited.*

Response: Thanks for raising up this question. The list of genes with high weights are provided in Table 3.

Rank	Segerstolpe	Xin	Fadista	Rank	Segerstolpe	Xin	Fadista
1	GCG	GCG	MALAT1	51	ITM2B	EIF4A2	RPS3A
2	TTR	MALAT1	EEF1A1	52	ENPP2	CTSD	RPL9
3	MALAT1	INS	TTR	53	ATP1A1	RBP4	SOD2
4	SERPINA1	TTR	FTH1	54	ANXA4	HNRNPH1	EIF4B
5	SPP1	FTL	GCG	55	HNRNPH1	BSG	HSPA8
6	B2M	PPP1CB	CPE	56	ALDOB	EEF2	PKM
7	FTH1	PCSK1N	GNAS	57	CD164	RPS3	SCG2
8	CHGA	CHGB	RPL4	58	HLA-A	PDK4	RPS24
9	PIGR	PSAP	APP	59	RIN2	SSR1	CD74
10	IAPP	CHGA	CTSD	60	ASAHI	SCD	SQSTM1
11	SST	EGR1	HSP90AA1	61	TMSB10	DNAJC3	TMBIM6
12	FTL	SRSF6	RPLP0	62	BSG	SAR1A	TXNRD1
13	CALM2	FTH1	RPL7A	63	CLDN4	GPX4	LCN2
14	CHGB	HSP90AB1	HSP90AB1	64	TMEM59	PLD3	RPL14
15	SERPINA3	SPINT2	HSP90B1	65	PPY	ATP6AP1	PDIA3
16	ACTG1	MAP1B	UBC	66	C10orf10	ANP32E	HDLBP
17	SCG5	RIN2	CANX	67	HSPA8	TBL1XR1	HNRNPK
18	ALDH1A1	GNAS	PAM	68	REG1B	GNB2L1	SCARB2
19	TM4SF4	SCG5	RPS6	69	P4HB	SLC22A17	RPL13A
20	REG3A	CSNK1A1	SERPINA3	70	LCN2	PAFAH1B2	LINC00657
21	GAPDH	PTEN	EIF4G2	71	PKM	RTN4	DSP
22	PPP1CB	TSPYL1	RPS4X	72	ATP6V0B	TMED4	SPINT2
23	ACTB	C6orf62	HSPA5	73	PSAP	CST3	REG1B
24	PRSS1	RPL3	ITGB1	74	LRRC75A-AS1	CD63	HNRNPC
25	RBP4	DPYSL2	IAPP	75	S100A11	TOB1	RPL15
26	GDF15	UBC	TPT1	76	MUC13	HLA-A	ENO1
27	COX8A	SCG2	RPL5	77	MAP1B	CLU	RPS11
28	ALDOA	ALDH1A1	SLC7A2	78	CD59	TTC3	GANAB
29	PDK4	PFKFB2	HNRNPA1	79	SLC30A8	RPS11	CDH1
30	RPL8	CPE	ANXA2	80	CPE	G6PC2	PEG10
31	H3F3B	C10orf10	RPL7	81	CLPS	GRN	CLDN4
32	IGFBP7	TMBIM6	RPS18	82	CTSD	SERPINA1	GSTP1
33	S100A6	CRYBA2	PCSK1	83	ATP1B1	SSR4	TUBA1A
34	EEF2	FTX	ATP1A1	84	OLFM4	RPS6	RPS27A
35	TIMP1	HSPA8	IDS	85	TAGLN2	OAZ1	PRPF8
36	CFL1	HSP90AA1	GDF15	86	SCGN	MARCKS	HSPB1
37	GRN	H3F3B	RPS3	87	SERPING1	RPL15	RPS8
38	SPINT2	SLC30A8	RPSA	88	WFS1	SQSTM1	RPS12
39	SQSTM1	TLK1	CSDE1	89	LAPTM4A	RASD1	ACLY
40	KRT19	ETNK1	CLTC	90	TAAR5	DSP	MSN
41	CD63	B2M	RPL10	91	SLC22A17	COX8A	HNRNPA2B1
42	SLC40A1	DDX5	YWHAZ	92	RPL3	TIMP1	CTNNB1
43	G6PC2	FOS	RPL3	93	HERPUD1	ATP1B1	MORF4L1
44	REG1A	MAFB	SLC30A8	94	CD24	WFS1	SERINC1
45	DDX5	CD59	RPL6	95	CALR	PRDX3	KRT19
46	PCBP1	TM4SF4	TMSB10	96	CLDN7	CHP1	NCL
47	C6orf62	TMEM33	CD44	97	LAMP2	YWHAE	GPX4
48	CRYBA2	CAPZA1	NPM1	98	CST3	FAM46A	GNB1
49	CD74	CALM2	B2M	99	TMBIM6	RUFY3	RPS7
50	HLA-E	GPX3	PABPC1	100	CTSB	C4orf48	SEP2
	alpha	beta	delta		gamma	acinar	ductal

Table 3: Top 100 Genes with highest weights in the pancreatic islet analysis. The bulk/artificial bulk data are from Segerstolpe et al., Xin et al. and Fadista et al. and the single cell reference is obtained from 6 healthy subjects from Segerstolpe et al. This table is color-coded by well-known marker genes.

Rank	Beckerman	Craciun	Arvaniti	Rank	Beckerman	Craciun	Arvaniti
1	Kap	Malat1	Malat1	51	Cycs	Dbi	Rps14
2	mt-Atp6	Kap	Kap	52	Rplp1	Rps18	Cox4i1
3	Gpx3	mt-Atp6	Gpx3	53	Rpl23	Rps14	Rpl26
4	mt-Co1	Gpx3	S100g	54	Gatm	Cycs	Cox5a
5	mt-Cytb	mt-Co1	Ftl1	55	Rpl32	Cox4i1	Rps19
6	S100g	mt-Cytb	Fth1	56	Cyb5a	Uqcrb	Rpl10
7	mt-Co3	S100g	Rps29	57	Acsm2	Ndrgr1	Ttc36
8	mt-Co2	mt-Co3	Xist	58	Guca2b	Rpl10	Rpl35
9	mt-Nd4	mt-Co2	Rpl37a	59	Uqcrb	Rpl26	Gm8730
10	mt-Nd1	mt-Nd4	Rpl41	60	Rps14	Rps19	Dnase1
11	Ftl1	mt-Nd1	Fxyd2	61	Cox4i1	Acsm2	Itm2b
12	Fth1	Ftl1	Rpl38	62	Rpl26	Rpl35	Rpl35a
13	Rps29	Fth1	Rpl37	63	Cox5a	Cyb5a	Rps24
14	mt-Nd2	Rps29	Miox	64	Rps19	Miox	Gm10260
15	mt-Nd3	mt-Nd2	Eef1a1	65	Ttc36	Itm2b	Atp5l
16	Rpl37a	mt-Nd4l	Rpl39	66	Rpl10	Rpl35a	Slc34a1
17	Rpl41	mt-Nd3	Cox6c	67	Dnase1	Atp5l	Aldob
18	Fxyd2	Rpl37a	Rps28	68	Rpl35	Gm8730	Cela1
19	Rpl38	Rpl41	Rps27	69	Rpl35a	Akr1c21	Ass1
20	Rpl37	Xist	Cndp2	70	Atp5l	Rpl28	Prdx1
21	Miox	Fxyd2	Cyp4b1	71	Rps24	Slc34a1	Rpl28
22	Eef1a1	Rpl37	Ndufa4	72	Slc34a1	Prdx1	Rpl23a
23	Rpl39	Rpl38	Akr1c21	73	Gm8730	Aldob	Rpl6
24	Cox6c	Eef1a1	Atp1a1	74	Itm2b	Rps27a	Pck1
25	Rps28	Spink1	Acy3	75	Aldob	Cox6a1	Gm10709
26	mt-Nd5	Rpl39	Atp5k	76	Cela1	Rps24	2010107E04Rik
27	Rps27	Rps28	Cox7c	77	Ass1	Rpl23a	Cox6a1
28	Cndp2	Cox6c	Klk1	78	Prdx1	Rps4x	Slc25a5
29	Cyp4b1	Rps27	Ubb	79	Rpl28	Gm10709	Rps4x
30	Ndufa4	mt-Nd5	Atp5e	80	Rpl6	Slc25a5	Rps27a
31	Akr1c21	mt-Atp8	Rps2	81	Rpl23a	Ppia	Ldhb
32	Atp1a1	Atp1a1	Ndrgr1	82	Pck1	Cox5a	Cox6b1
33	Acy3	Cox7c	Rps23	83	2010107E04Rik	Rpl13	Rpl18a
34	Atp5k	Ubb	Gm10076	84	Cox6a1	Cox6b1	Calb1
35	Cox7c	Atp5e	Prdx5	85	Gm10709	Cox7a2	Rpl13
36	Klk1	Atp5k	Rps1	8 86	Slc25a5	Gatm	Atp5b
37	Atp5e	Ndufa4	Tpt1	87	Rps27a	Ass1	Rpl13a
38	Ubb	Rps2	Chchd10	88	Rps4x	Ndufa3	Cox7a2
39	Rps2	Rps23	Rplp0	89	Ldhb	Rpl18a	Ndufa3
40	Ndrgr1	Gm10076	Dbi	90	Cox6b1	Cyp4b1	Slc27a2
41	Rps23	Klk1	Rpl29	91	Calb1	Atp5j	Actb
42	Gm10076	Rps21	Rps21	92	Atp5b	Cox8a	Ppia
43	Prdx5	Rpl29	Rplp1	93	Cox7a2	Acy3	Rpl36a
44	Chchd10	Prdx5	Cycs	94	Rpl18a	Rpl36a	Atp5j
45	Tpt1	Rplp1	Rpl23	95	Ndufa3	Actb	Chpt1
46	Rps18	Tpt1	Rpl32	96	Slc27a2	Ndufa13	Rps15a
47	Dbi	Rpl23	Gatm	97	Rpl13	Rpl13a	Hrsp12
48	Rps21	Rpl32	Acsm2	98	Rpl36a	Ttc36	Ndufa13
49	Rplp0	Chchd10	Guca2b	99	Ppia	2010107E04Rik	Cox8a
50	Rpl29	Rplp0	Uqcrb	100	Atp5j	Gm10260	Ugt2b38
	PT	DCT	CD-IC		Pod0	T lymph	

Table 4: Top 100 genes with highest weights in the mouse kidney analysis in Step 1 of the tree-guided deconvolution procedure. The bulk/artificial bulk data are from Beckerman et al., Craciun et al. and Arvaniti et al. and the single cell reference is obtained by 7 healthy mice from Park et al. This table is color-coded by marker genes.

Immune	Rank	Beckerman	Craciun	Arvaniti	Rank	Beckerman	Craciun	Arvaniti
	1	Cd74	Apoe	Cd74	26	C1qb	Npc2	C1qb
	2	Lyz2	S100a6	Lyz2	27	Nkg7	Gzma	Nkg7
	3	Ccl5	S100a4	Ccl5	28	Ccl4	Capza2	Vim
	4	H2-Aa	Psap	H2-Aa	29	Vim	Ly6e	Ccl4
	5	H2-Ab1	Nkg7	H2-Ab1	30	Ly6c2	Ly6c2	Ly6c2
	6	Tmsb10	Crip1	Tmsb10	31	Ms4a4b	Serinc3	Ms4a4b
	7	Gzma	Cd3g	Gzma	32	Sat1	Fos	Sat1
	8	H2-Eb1	Ccl3	H2-Eb1	33	C1qc	Pou2f2	C1qc
	9	Plac8	Ccnd2	Plac8	34	S100a10	Ctsz	S100a10
	10	Cst3	Slpi	Cst3	35	H3f3a	Cd74	H3f3a
	11	Ifi272a	Gm2a	Ifi272a	36	Ctss	Il7r	Ctss
	12	Slpi	Ssr4	Slpi	37	Gngt2	H2afy	Gngt2
	13	Ifitm3	Lck	Ifitm3	38	S100a6	Ctsb	S100a6
	14	Apoe	Spi1	Apoe	39	S100a4	Ifngr1	S100a4
	15	Tyrobp	Fxyd5	Tyrobp	40	Lst1	Tgfb1	Lst1
	16	Actg1	Ccl4	Actg1	41	Klf2	Sub1	Klf2
	17	Crip1	Gzmb	Crip1	42	Msrb1	Socs2	Msrb1
	18	Fcer1g	Cnn2	Fcer1g	43	H2afz	Ifitm3	H2afz
	19	Cebpb	Id2	Cebpb	44	Wfdc17	Itgb7	Wfdc17
	20	C1qa	Cybb	C1qa	45	Arpc1b	Cd79a	Arpc1b
	21	AW112010	Sep1	AW112010	46	Ifitm2	Ltb	Ltb
	22	Ly6e	Hsp90b1	Ly6e	47	Ltb	Fyb	Ifitm2
	23	Id2	Itgb2	Id2	48	S100a11	Tspan32	S100a11
	24	Psap	Ccl6	Psap	49	Lgals3	Sat1	Mzb1
	25	Lgals1	Lsp1	Lgals1	50	Mzb1	Xbp1	Lgals3
Epithelial	Rank	Beckerman	Craciun	Arvaniti	Rank	Beckerman	Craciun	Arvaniti
	1	Hbb-bs	Hbb-bs	Hbb-bs	26	Gm5424	Slc12a3	Slc22a28
	2	Hba-a1	Hba-a1	Hba-a1	27	Slc12a3	Slc22a28	Slc22a29
	3	Umod	Slco1a1	Slco1a1	28	Nrp1	Slc22a29	Emcn
	4	Slco1a1	Slc22a6	Slc22a6	29	Igfbp5	Ly6c1	Car12
	5	Slc22a6	Pvalb	Nat8	30	Ehd3	Car12	Aspdh
	6	Pvalb	Nat8	Pvalb	31	Slc22a28	Aspdh	Akr1c14
	7	Nat8	Umod	Mep1a	32	Slc12a1	Igfbp5	Ly6c1
	8	Mep1a	Mep1a	Umod	33	Slc22a29	Akr1c14	Hexb
	9	Egf	Slco1a6	Slco1a6	34	Car12	Atp6v1g3	BC035947
	10	Slco1a6	Ces1f	Ces1f	35	Aspdh	Ehd3	Igfbp5
	11	Ces1f	Hbb-bt	Hbb-bt	36	Akr1c14	Hexb	Atp6v1g3
	12	Hbb-bt	Egf	Snhg11	37	Kdr	Slc12a1	Nrp1
	13	Snhg11	Snhg11	Tmigd1	38	Atp6v1g3	BC035947	Slc13a1
	14	Tmigd1	Tmigd1	Egf	39	Hsd11b2	Slc13a1	Slc12a1
	15	Acsm3	Acsm3	Acsm3	40	Hexb	Col6a6	Col6a6
	16	Slc22a30	Slc22a30	Slc22a30	41	Eng	Gm4450	Gm4450
	17	Gm11128	Cyp2a4	Gm11128	42	BC035947	Kdr	Adamts15
	18	Aqp2	Hba-a2	Cyp2a4	43	Pi16	Adamts15	Ehd3
	19	Cyp2a4	Aqp2	Hba-a2	44	Slc13a1	Hsd11b2	Aspa
	20	Fxyd4	Aqp1	Gm5424	45	Col6a6	Aspa	Mogat1
	21	Emcn	Gm5424	Slc17a1	46	Gm4450	Apela	D630029K05Rik
	22	Aqp1	Slc17a1	Aqp1	47	Egfl7	Mogat1	Gm15638
	23	Hba-a2	Plpp1	Aqp2	48	Adamts15	D630029K05Rik	Hsd11b2
	24	Ly6c1	Fxyd4	Slc12a3	49	Meis2	Eng	Akr1c18
	25	Slc17a1	Emcn	Fxyd4	50	Aspa	Gm15638	Smlr1
		PT	DCT	CD-IC	LOH	CD-PC	Endo	Podo
		Neutro	T lymph	Macro	Fib	B lymph	NK	

Table 5: Top 100 genes with highest weights in the mouse kidney analysis in Step 2 of the tree-guided deconvolution procedure (separated by epithelial and immune cells). The bulk data are from Beckerman et al., Craciun et al. and Arvaniti et al. and the single cell reference is obtained by 7 healthy mice from Park et al. This table is color-coded by marker genes.

MuSiC selected well-known marker genes from deconvolution, highlighted with colors. However, some of the high-weight genes are not necessarily marker genes. We emphasize that the cross-subject variation and cross-cell-type variation are very different concepts. The cross-subject variation measures the consistence of genes across subjects while the cross-cell-type variation measures the cell type specificity of genes. We examine further on those high-weight non-markers genes, we found that those genes tend to be consistently expressed across subjects, and are usually highly expressed genes. Although they are not exclusively expressed in a certain cell type hence not marker genes, they are differentially expressed across cell types, thus offering power to differentiate different cell types. We believe that MuSiC benefit from those genes and hence yield accurate cell type proportions than methods that only use marker genes in deconvolution.

We added Table 3-5 to the Supplementary Information as Supplementary Table 5-7 and the discussion of weight on page 11, line 412-425.

2. *It is not clear how well the different measures of mRNA abundance interconvert, e.g. TPM, RPKM, drop-seq counts.*

Response: Thanks for bringing up this problem. Below, we provide explanation of each measure and how different measures are connected. Drop-seq counts, as we understood, are UMI read counts, which does not require interconversion and can be treated as read counts in MuSiC deconvolution.

MuSiC links bulk and single-cell gene expression by mRNA molecule counts. There are many measures of mRNA abundance, such as read counts, UMI counts, RPKM and TPM. As molecule counts are not observed in real studies, we approximate the molecule counts by read counts and estimate cell type proportions based on **assumptions (A1-A3) in the revised maintext**. The interconversion between other gene expression measures and read count determines if MuSiC can utilize other measures as the input for deconvolution. One step in MuSiC estimation is the use of average library size as a proportional measure of average cell size for a given cell type, which is absent in normalized measurements of mRNA abundance such as RPKM and TPM. For RPKM, we would need the average library size for each cell type to be provided, or the average cell size for each cell type to be obtained from other sources. Cell type proportions cannot be estimated by MuSiC with TPM information alone. Below, we derive the relationships of various types of gene expression measures in detail.

Let L_g denote the length of gene g , and the corresponding RPKMs of bulk and single-cell data are denoted by \widehat{X}_{jg} and \widehat{X}_{jgc} , respectively. For simplicity, we omit the 10^3 scaler for now. By definition,

$$\widehat{X}_{jg} = \frac{\widetilde{X}_{jg}/L_g}{\sum_{g'=1}^G \widetilde{X}_{jg'}}, \quad \widehat{X}_{jgc} = \frac{\widetilde{X}_{jgc}/L_g}{\sum_{g'=1}^G \widetilde{X}_{jg'c}}, \quad (4)$$

where \widetilde{X}_{jg} and \widetilde{X}_{jgc} denote the bulk and single cell read counts, respectively.

Based on the model set-up described earlier, we can show that the relationship between bulk and single-cell RPKM is We can show that

$$\widehat{X}_{jg} \propto \frac{\widetilde{X}_{jg}}{L_g} = \sum_{k=1}^K \sum_{c \in C_j^k} \left(\frac{\widetilde{X}_{jgc}/L_g}{\sum_{g'=1}^G \widetilde{X}_{jg'c}} \cdot \sum_{g'=1}^G \widetilde{X}_{jg'c} \right) = \sum_{k=1}^K \sum_{c \in C_j^k} \widehat{X}_{jgc} \widetilde{S}_{jc}, \quad (5)$$

where \widetilde{S}_{jc} is the library size of cell c . Equation (5) can be further approximated by

$$\widehat{X}_{jg} \propto \sum_{k=1}^K \sum_{c \in C_j^k} \widehat{X}_{jgc} \widetilde{S}_{jc} \approx \sum_{k=1}^K m_j^k \widehat{\theta}_{jg}^k \widetilde{S}_j^k, \quad (6)$$

where $\widehat{\theta}_{jg}^k = \frac{\sum_{c \in C_j^k} \widetilde{X}_{jgc}}{m_j^k}$ is the average RPKM of gene g in subject j for cell type k .

To utilize multi-subject information, we assume $\widehat{\theta}_{jg}^k$ follows the same assumption as (A1), that is, individuals with scRNA-seq and bulk RNA-seq are from the same population, with their cell-type specific average RPKM $\widehat{\theta}_{jg}^k$ following the same distribution with mean $\widehat{\theta}_g^k$ and variance $\widehat{\sigma}_{gk}^2$,

$$\widehat{\theta}_{jg}^k \sim \widetilde{F}(\widehat{\theta}_g^k, \widehat{\sigma}_{jg}^2).$$

Assumption (A2) states that the ratio of average library size is consistent across subjects and studies, which justified the use of \widetilde{S}_j^k from other studies if there quantities are no available for the same data set. The linear relation between bulk RPKM and average cell-type specific single cell RPKM is approximated by formula (6). Since this is an approximation, MuSiC estimates using RPKM may not be as accurate as those using read or UMI count. In our test of MuSiC using RPKM values for the pancreatic islets bulk mixture experiment, we found that it is not as accurate as MuSiC estimates using read count, but still higher than NNLS, BSEQ-sc, and Cibersort (Figure 6).

Figure 6: The Heatmap of MuSiC deconvolution with RPKM as the input. Single cell and bulk data are from Segerstolpe et al. and the ratio of library size are borrowed from Segerstolpe et al. read counts.

Another widely used normalized mRNA measure is TPM. let \widehat{Z}_{jg} and \widehat{Z}_{jgc} denote the bulk and single-cell TPM values, respectively. By definition,

$$\widehat{Z}_{jg} = \frac{\widetilde{X}_{jg}/L_g}{\sum_{g'=1}^G \widetilde{X}_{jg'}/L_{g'}}, \quad \widehat{Z}_{jgc} = \frac{\widetilde{X}_{jgc}/L_g}{\sum_{g'=1}^G \widetilde{X}_{jg'c}/L_{g'}}. \quad (7)$$

Let Z_{jg} and Z_{jgc} be the gene length normalized read counts in bulk and single cell, that is, $Z_{jg} = \widetilde{X}_{jg}/L_g$ and $Z_{jgc} = \widetilde{X}_{jgc}/L_g$. The link between bulk and single-cell TPM is

$$\widehat{Z}_{jg} \propto Z_{jg} = \sum_{k=1}^K \sum_{c \in C_j^k} Z_{jgc} = \sum_{k=1}^K \sum_{c \in C_j^k} \left(\frac{Z_{jgc}}{\sum_{g'=1}^G Z_{jg'c}} \cdot \sum_{g'=1}^G Z_{jg'c} \right) = \sum_{k=1}^K \sum_{c \in C_j^k} \widehat{Z}_{jgc} \widehat{S}_{jc}, \quad (8)$$

where \widehat{S}_{jc} is the summation of normalized read counts in cell c for subject j .

Equation (8) suggests that it is difficult to make assumptions or approximation to express relative abundance as a function of TPM.

We have put the interconversion discussion in the **Discussion** (on page 7, line 275-280) and **Method** section: Interconversion of different gene expression measurements (on page 12-14, line 471-519). We also added the example of RPKM deconvolution to **Supplementary Figure 8d**.

3. *There is a danger of overfitting when running the model. Authors should add additional sensitivity analysis to assure reader that the cell number estimates are not sensitive to small changes in assumptions.*

Response: Thanks for raising this question. To address the reviewer's concern, we have conduct additional sensitivity analysis below.

Our method makes two assumptions:

(A1) the relative abundance θ_{jg}^k follows the distribution with mean θ_g^k and variance σ_{gk}^2 .

We note that we do not assume the distribution to be of any functional form, therefore, this assumption generally holds.

(A2) the ratio of average cell size S_j^k across cell types are the same across different subjects

$$\frac{S_j^k}{S_{j'}^{k'}} = \frac{S_{j'}^k}{S_{j'}^{k'}}, \quad \text{for all } j, j' \in \{1, \dots, N\} \text{ and } k, k' \in \{1, \dots, K\}. \quad (9)$$

To show that this assumption is reasonable, Figure 7 shows the library size ratio (relative to alpha cells) of single cell data from Segerstolpe et al.

Figure 7: Library size ratio of the 6 major cell types in pancreatic islet. Single cell data is from Segerstolpe et al. The ratio is calculated as the library size of a cell type as the library size of the alpha cell library size. The errorbar represents S.E. of estimated library size ratio across subjects.

Figure 7 shows the library ratio (relative to alpha) of 6 major cell types from Segerstolpe et al. The errorbars show that the ratio is not strictly the same and assumption (A2) is not strictly held. Even though, MuSiC provides accurate estimations and is not sensitive to assumption (A2). The numerical bias analysis are provided in Response to Reviewer 1, Comment 1.

We added more explanation of assumption (A2) in the **revised maintext** on page 8, line 331-333.

4. *scRNA-Seq data are highly variable for a given cell type and tend not to be normally distributed. Were mean values used, would it come out the same if medians were used.*

Response: In MuSiC, the bulk and single cell data are linked together based on relationship derived in **Equation (1-3) in the maintext**. This relationship is based on cross-subject mean and variance. To improve robustness, we can either use trimmed mean in MuSiC, or apply median to our method through the use of quantile regression rather than ordinary regression.

5. *DCT is far from the second most abundant epithelial cell type in kidney. Thick ascending limb cells are at least three times more plentiful. Authors ignore their own warning that cell isolation for scRNA-Seq is biased toward certain cell type.*

Response: Thanks for your note. In our reading of the literature the true cell composition of the kidney is yet to be defined. We agree that the loop of Henle is the longest segment of the kidney, but cell number and segment length will not necessarily fully correlate.

While single cell sequencing has been paradigm shifting improving our understanding of cell types, all currently available methods have significant limitations including cell drop-out. Therefore single cell methods cannot fully recapitulate the composition of solid organs as some cell types are not covered well. The loop of Henle and some other cells that presumed to reside in the renal medulla were not covered well in the large dataset generated by Park et al. The kidney is a highly complex organ with multiple highly different cell types, once larger and better datasets are generated MuSiC can be rerun to refine the cell composition in the kidney. Another possibility could be that the whole kidney dataset we downloaded from GEO did not cover the kidney medulla and the loop of Henle cell well, reflected by our cell composition analysis. We acknowledge this bias of scRNAseq.

On the other hand, we would like to emphasize that our method has performed extremely well, even in a tissue with more than a dozen different cell types.

6. *The authors' conclusion that DCT is similar to PT does not measure up to knowledge about the structure and function of these two cell types. This is based on dichotomous clustering that does not really tell how close they are. K-means clustering should be better since cell types are pre-specified. Also, for the reader it would be good to pull out the transcript values that are the basis of this classification*

Response: Thanks for raising this question. First, we want to clarify that the similarity is a relative concept, that is, when compared to immune cells, PT and DCT cells are more similar to each other. Also, the similarity measures the gene expression of DCT and PT cells, not their structure nor function. The similarity of gene expression leads to collinearity issue in regression. The main purpose of the recursive tree procedure is not to determine clusters, but rather to get rid of collinearity in regression. We also did further analysis of gene expression similarity without mitochondrial genes. The mitochondrial free similarity analysis is shown in Figure 8.

Figure 8: Clustering of mouse single cell data by design matrix, which is generated without mitochondrial genes.

Without mitochondrial genes, DCT and PT cells are still the most similar in design matrix and cross-subject variation. Although, DCT and PT are not similar in structure and function, they share similar pattern in gene expressions. The similarity in gene expression require extra efforts to distinguish when deconvolving with expression.

7. *The authors coin a term 'stable' to describe variability. An explicit definition is needed. This is probably a bad term because biologist that investigate mRNA abundance regulation use the term 'stability' to refer to mRNA half life.*

Response: Thanks for your suggestion. To avoid confusion, we have changed 'stable' to 'consistent' in the

Maintext and Supplementary Information. We define consistent genes to be these showing low variable genes across subjects.

8. *The analysis of the rat data from Lee et al. describes the measurements as “bulk rat RNA-seq data”. Although the Lee samples are multicellular, they are made up of a single cell type. So Lee’s DCT samples were already 100 percent DCT cells. Authors should remove the word ‘bulk’ and explain that these are essentially heterogeneous samples. The interpretation is stated as, “knowledge about the dominant cell type at its mapped position, e.g. DCT cells come from the DCT region”. ‘Region’ is the wrong word here. ‘Segment’ is more appropriate for microdissected tubules.*

Response: Follow the reviewer’s suggestion, we have changed ‘bulk’ to ‘microdissected aggregated’ and ‘Region’ to ‘Segment’ in the maintext and in the Supplementary Information.

Response to Reviewer #3

This manuscript presents a new statistical method and an open source R package call MuSiC, to identify and estimate the proportion of individual cell types in bulk RNA-seq sample, using multiple single-cell RNA-seq data sets as a reference. The key innovative ideas in this methodology include: (1) using cross-subject and cross-cell stability as a measure to identify good cell-type specific markers; and (2) a recursive tree-guided deconvolution scheme, which is helpful in discovering and estimating low-frequency cell types. The authors have shown the applicability of their method using two case studies - pancreatic islets in humans, and a cross-species analysis of kidney cells. The paper is well written, the method is logical and clearly presented, and the experiment results seem solid. I like the authors’ idea of using scRNA-seq data as reference for deconvolution of bulk RNA-seq data. This paper clearly contains methodological innovation.

Thanks for your encouragements!

Nonetheless, to fully evaluate the real-life applicability of MuSiC, I have the following questions:

1. *scRNA-seq data are known to contains a high proportion of signal dropouts and measurement variability. I suspect it is difficult to use multiple scRNA-seq data with different dropout rate as a reference. Do the authors have any experimental data (using simulation data, for example) to show that MuSiC is robust to different level of noise in the scRNA-seq data?*

Response: This is a good suggestion. In the revision, we simulated scRNA-seq reference with different dropout rates from real single cell dataset. As in Figure 2a and b from the main text, the artificial bulk data is constructed from single cell data from Segerstolpe et al. and Xin et al. The reference scRNA-seq data is simulated by adding noise to Segerstolpe et al. single cell data, with dropout rate π_{gc} is generated following the model in Jia et al. (2017) [1],

$$\pi_{gc} = \frac{1}{1 + \exp\{k \ln(X_{gc})\}}, \quad (10)$$

where X_{gc} is the observed read counts, k is the dropout rate parameter. The simulated read count X'_{cg} is generated by

$$P(X'_{cg} = X_{cg}) = \pi_{gc}, \quad P(X'_{cg} = 0) = 1 - \pi_{gc}. \quad (11)$$

We studied 4 different dropout rates with $k = 1, 0.5, 0.2, 0.1$, with smaller k leading to higher dropout rate. The evaluation of Segerstolpe et al. constructed artificial bulk data are shown by Figure 9 and Table 6; the evaluation of Xin et al. constructed artificial bulk data are shown by Figure 10 and Table 7.

Figure 9: Heatmap of real and estimated cell type proportions. Artificial bulk data is constructed by Segerstolpe et al. while the single cell reference is obtained from 6 healthy subject from Sgerstolpe et al. with different dropout rate: $k = 1, 0.5, 0.2, 0.1$.

Figure 10: Heatmap of real and estimated cell type proportions. Artificial bulk data is constructed by Xin et al. while the single cell reference is obtained from 6 healthy subject from Segerstolpe et al. with different dropout rate: $k = 1, 0.5, 0.2, 0.1$.

Dropout	RMSD	mAD	R
$k = 1$	0.040	0.030	0.973
$k = 0.5$	0.042	0.031	0.970
$k = 0.2$	0.051	0.039	0.955
$k = 0.1$	0.050	0.037	0.957

Table 6: Evaluation of estimated cell type proportions. Artificial bulk data is constructed by Segerstolpe et al. while the single cell reference is obtained from 6 healthy subject from Sgerstolpe et al. with different dropout rate: $k = 1, 0.5, 0.2, 0.1$.

Dropout	RMSD	mAD	R
$k = 1$	0.100	0.064	0.936
$k = 0.5$	0.105	0.067	0.925
$k = 0.2$	0.123	0.078	0.896
$k = 0.1$	0.123	0.080	0.895

Table 7: Evaluation of estimated cell type proportions. Artificial bulk data is constructed by Xin et al. while the single cell reference is obtained from 6 healthy subject from Sgerstolpe et al. with different dropout rate: $k = 1, 0.5, 0.2, 0.1$.

From the evaluations above, we found that in general, adding more dropout noise leads to lower MuSiC estimation accuracy, but the effect is quite small. Comparing with evaluations in **Figure 2 in the maintext**, MuSiC attained consistently higher accuracy than current methods even with dropout noise.

We have added the robustness analysis of dropout in the **revised maintext** on page 6, line 251-254 and **Supplementary Note 5**. We also added Figures 9-10 as **Supplementary Figure 8a-b**.

2. *Have you considered the impact of normalization and batch effect correction to the performance of MuSiC?*

Response: Yes. There are two key factors in MuSiC model, the cell type specific cross-subject mean and cross-subject variance of relative abundance θ_{jg}^k . When batch effect is present, the variance of relative abundance will generally increase for all cell types. This means that the batch effect will be absorbed in σ_{kg} , meaning that MuSiC not only up-weights cross-subject consistent genes, but also cross-batch consistent genes.

Thus, by down-weighting cross-batch variable genes, MuSiC effectively deals with batch effects. Batch effects may be more of a problem for existing methods that only rely on the mean values and ignores cross sample variation.

We have added the discussion of batch effect in **the revised maintext** on page 10-11, lines 405-410.

3. *The detail for performing the cross-species analysis (end of page 7) was missing. Did the authors assume there is a 1-1 mapping of homologous genes between two species? How did they deal with complex homology relationships? I suspect the claim about 'cross-species applicability' probably only extend to mammals, but not other organisms that are further away in the evolutionary tree.*

Response: You are right. The 'cross-species applicability' only extends to similar species such as mouse and rat, where we only use homologous genes between mouse and rat. We have not yet conducted a detailed study of bulk RNA-seq deconvolution using reference data derived from a more distant species. It would be interesting, and a big undertaking, to figure out how distant the species can be for it to work. We feel that is out of the scope of this paper, and so we simply stressed in the revision that we have only tried very closely related species.

We have added more descriptions on the cross-species analysis in **the revised maintext** on page 5, line 223.

References

- [1] Jia, C., Hu, Y., Kelly, D., Kim, J., Li, M., & Zhang, N. R. (2017). Accounting for technical noise in differential expression analysis of single-cell RNA sequencing data. *Nucleic acids research*, 45(19), 10978-10988.

REVIEWERS' COMMENTS:

Reviewer #1 (Remarks to the Author):

I am happy to say that the authors have addressed the majority of my concerns. I have only two minor comments:

- The authors seem to have misunderstood my comment #3 about Equation 8. Equation 3 defines $Y_{\{jg\}}$ to be bounded in $[0, 1]$; if the error in Equation 8 is normally distributed, it would be possible to have a non-zero probability mass outside the range of possible values for $Y_{\{jg\}}$. The authors mention C_j but this is a constant value and will not provide any protection from impossible values. Now, as I mentioned before, this might not be a problem if the variance of the error is so low that the bounds on $Y_{\{jg\}}$ do not matter, i.e., the probability mass outside of the range is negligible. However, it would be best to be explicit about this; or the authors should consider error distributions that are more conducive to the bounds, e.g., a Gamma distribution via `statmod::glmGamFit`.

- For future reference, the voom mean-variance relationship is typically observed in the variance of the log-expression values against the log-mean for filtered bulk RNA-seq data. Without filtering, the curve actually dips down to zero - see, for example, Figure 8 in <https://bioconductor.org/packages/release/workflows/vignettes/singleCellInst/doc/work-1-reads.html>. Of course, for the purposes of deconvolution, there is no need to be constrained to this mean-variance relationship; it is simple enough to empirically model the relationship from the data.

Reviewer #2 (Remarks to the Author):

Good job responding to recommendations. Software offers a useful new tool that will likely be widely utilized.

Reviewer #3 (Remarks to the Author):

The authors have adequately addressed all my concerns. I am pleased to see the significant improvement the authors have made in this revision. I think this will be a useful tool for the bioinformatics community. I hope the authors will commit to maintaining the R package in the future.

Minor comment:

line 408: 'no only' -> 'not only'

Response to Reviewer #1

I am happy to say that the authors have addressed the majority of my concerns. I have only two minor comments:

1. *The authors seem to have misunderstood my comment #3 about Equation 8. Equation 3 defines Y_{jg} to be bounded in $[0, 1]$; if the error in Equation 8 is normally distributed, it would be possible to have a non-zero probability mass outside the range of possible values for Y_{jg} . The authors mention C_j but this is a constant value and will not provide any protection from impossible values. Now, as I mentioned before, this might not be a problem if the variance of the error is so low that the bounds on Y_{jg} do not matter, i.e., the probability mass outside of the range is negligible. However, it would be best to be explicit about this; or the authors should consider error distributions that are more conducive to the bounds, e.g., a Gamma distribution via `statmod::glmGamFit`.*

Response: Thanks for your further explanation of this problem. We agree that a more complex model, such as Gamma, may more accurately describe the data, but in practice, the normal distribution is often used for bounded value and gives acceptable performance. Here we rely on this mostly for simplicity and computational convenience. In the Supplementary, we added supplementary note 7 in this regard.

2. *For future reference, the voom mean-variance relationship is typically observed in the variance of the log-expression values against the log-mean for filtered bulk RNA-seq data. Without filtering, the curve actually dips down to zero - see, for example, Figure 8 in <https://bioconductor.org/packages/release/workflows/vignettes/simpleSingleCell/inst/doc/work-1-reads.html>. Of course, for the purposes of deconvolution, there is no need to be constrained to this mean-variance relationship; it is simple enough to empirically model the relationship from the data.*

Response: Thanks for recommend this paper. We have checked the mean-variance relationship of existing in our previous response, and will certainly consider the constraints for future investigation.

Reviewer #2's comments

Good job responding to recommendations. Software offers a useful new tool that will likely be widely utilized.

Response to Reviewer #3

The authors have adequately addressed all my concerns. I am pleased to see the significant improvement the authors have made in this revision. I think this will be a useful tool for the bioinformatics community. I hope the authors will commit to maintaining the R package in the future.

Minor comment: line 408: 'no only' → 'not only' Thanks for pointing out this typo. We have fixed the typo in line 408.